# Potential for bias in effective climate sensitivity from state-dependent energetic imbalance

**Benjamin M. Sanderson**[1] **and Maria Rugenstein**[2]

[1]CICERO, Oslo, Norway
[2]Colorado State University, Fort Collins CO, USA

**Correspondence:** Benjamin Sanderson (benjamin.sanderson@cicero.oslo.no)

**Abstract.** To estimate equilibrium climate sensitivity from a simulation where a step change in carbon dioxide concentrations is imposed, a common approach is to linearly extrapolate temperatures as a function of top of atmosphere energetic imbalance to estimate the equilibrium state ("Effective Climate Sensitivity"). In this study, we find that this estimate may be biased in some models due to state-dependent energetic leaks. Using an ensemble of multi-millennial simulations of climate model response to a constant forcing, we estimate equilibrium climate sensitivity through Bayesian calibration of simple climate models which allow for responses from subdecadal to multi-millennial timescales. Results suggest potential biases in Effective Climate Sensitivity in the case of particular models where radiative tendencies imply energetic imbalances which differ between pre-industrial and quadrupled $CO_2$ states, whereas for other models even multi-thousand year experiments are insufficient to predict the equilibrium state. These biases draw into question the utility of effective climate sensitivity as a metric of warming response to greenhouse gases and underline the requirement for operational climate sensitivity experiments on millennial timescales to better understand committed warming following a stabilisation of greenhouse gases.

## 1 Introduction

Equilibrium Climate Sensitivity (*ECS*) is the theoretical equilibrium increase in global mean temperature experienced in response to an instantaneous doubling in Earth's carbon dioxide concentrations over pre-industrial levels. Introduced as a metric of response of the Earth System to greenhouse gases in the early years of computational climate science (Charney et al., 1979; Hansen et al., 1984), it remains a very common metric of the sensitivity of the Earth to greenhouse gas forcing (Knutti et al., 2017; Masson-Delmotte et al., 2021).

Measuring *ECS* in a coupled climate model, however, is difficult owing to the time required for the equilibration of the system to a change in forcing (Wetherald et al., 2001; Solomon et al., 2010; Jarvis and Li, 2011) necessitating simulations of multiple millennia to obtain a near-equilibrated estimate of temperature response (Rugenstein et al., 2020). The computational burden of conducting such simulations implies that standard practise for model assessment is to measure an "Effective Climate Sensitivity" (*EffCS*) using feedbacks extrapolated from those simulated in the first 150 years simulation forced with a step-wise quadrupling of $CO_2$ (Gregory et al., 2004; Murphy, 1995; IPCC, 2013; Forster, 2016; Andrews et al., 2012).

A core assumption in the calculation of *EffCS* is that the system will ultimately stabilise in a state of energetic balance (Gregory et al., 2004). However, in practise a number of models exhibit energetic radiative top of atmosphere imbalances in the control state in both CMIP5 (Hobbs et al., 2016) and CMIP6 (Irving et al., 2021), and as such the Effective Climate Sensitivity is calculated using net flux anomalies relative to the control mean top of atmosphere net radiative fluxes. However, it remains untested as to whether such models will ultimately converge to the same state of imbalance.

In the present study, we consider an alternative approach for calculating climate sensitivity from a climate simulation in which there is a step change in carbon dioxide concentrations. We consider how the method of calculating effective climate sensitivity, either from initial response or from millennial scale simulations, may be potentially subject to biases arising from assumptions on the equilibrated radiative state. Finally, we consider how these uncertainties relate to our confidence in the relationship between transient and equilibrium climate feedbacks.

We consider the role of non-equilibrated models in the context of recent research, which has highlighted potential uncertainties in the *EffCS* approximation of *ECS* - studies have found that net radiative feedbacks can exhibit both timescale and state dependencies (e.g., Senior and Mitchell 2000; Armour et al. 2013; Andrews et al. 2015; Rugenstein et al. 2016; Proistosescu and Huybers 2017; Pfister and Stocker 2017; Dunne et al. 2020; Andrews et al. 2018; Bloch-Johnson et al. 2021) both of which draw into question the implicit constant feedback assumption used to calculate *EffCS*.

The LongrunMIP project set out in part to quantify this error by running a subset of ESMs in idealised carbon dioxide perturbation experiments with simulations of millennial timescale response (Rugenstein et al., 2019). Initial studies compared the *EffCS* as derived using the first 150 years of the simulation with that derived using the last 15 percent of warming in multi-thousand year experiments - finding that the accuracy of the *EffCS* varied by model, but the two methods differed by 5-37% in the estimate of *ECS* (Rugenstein et al., 2020). A follow-up study (Rugenstein and Armour, 2021) considered a range of approaches for characterising feedbacks on different timescales, and found that feedbacks assessed in the period 100-400 years after the initial quadrupling of $CO_2$ concentrations may provide a practical prediction of equilibrium response accurate within 5% or less. They found also, however, there were large inconsistencies in some models between estimates of climate sensitivity derived from extrapolation to radiative equilibrium and those methods which relied on a fitting of exponentially decaying temperature trend, leaving uncertainty on the best practise for integrating model-derived *EffCS* distributions into uncertainty in long term warming trajectories.

A general assessment of the likely range of *EffCS* (Sherwood et al., 2020) (which itself informed the Forster et al. (2021) assessed likely *EffCS* range) rested strongly on combined historical and paleo evidence, contributing to the headline result that values of *EffCS* of greater than 4.7K are unlikely. These findings somewhat challenge the use of the CMIP6 ensemble of climate models as a proxy for climate projection uncertainty in assessment, given approximately 1/3 of the ensemble have apparent *EffCS* values of greater than 4.7K (O'Neill et al., 2016; Eyring et al., 2016; Meehl et al., 2020; Zelinka et al., 2020) - leading to arguments that such 'hot models' should be excluded from assessment (Hausfather et al., 2022).

So can these models be ruled out? Although studies suggest that post-1980 warming may help constrain the Transient Climate Response (Jiménez-de-la Cuesta and Mauritsen, 2019; Nijsse et al., 2020; Tokarska et al., 2020), recent historical warming alone is only weakly correlated with *EffCS* in the CMIP5 and CMIP6 ensemble (Tokarska and Gillett, 2018). In the present study, we find that this might in part be due to the fact that a key assumption in *EffCS* (that the model will return to the radiative balance observed in the control simulation) may not hold in a number of CMIP-class models.

## 2  Methods

We consider fits of a simple multi-timescale model to idealised climate change experiments from LongRunMIP (Rugenstein et al., 2019), which provide in general an estimate of the multi-millennial response of the Earth System to a constant radiative forcing level. The supplementary material also illustrates results from CMIP5 (Taylor et al., 2012) and CMIP6 (Eyring et al., 2016), but in general these simulations are insufficiently long to constrain the simple model response.

We assume that the temperature and radiative response to a step change in forcing can be modelled by a sum of exponential decay terms, a basis set which is consistent with the general solution of two layer simple climate models and one which holds for the solution of a number of proposed multi-layer linear energy balance models in response to constant forcings (Caldeira and Myhrvold, 2013; Proistosescu and Huybers, 2017; Sanderson, 2020; Geoffroy et al., 2013a; Winton et al., 2010; Smith et al., 2018; Geoffroy et al., 2013b) . It has been shown also that some non-linear models have a solution set which can also be expressed in the same exponential basis (Proistosescu and Huybers, 2017; Bastiaansen et al., 2021). We consider $N$ exponential response modes, such that:

$$T_p(t) = \sum_{n=1}^{N} S_n(1 - e^{-(t/\tau_n)}) + T_0 \tag{1a}$$

$$R_p(t) = \sum_{n=1}^{N} R_n(-e^{-(t/\tau_n)}) + R_{extrap}^{4x}, \tag{1b}$$

where $T_p(t)$ and $R_p(t)$ are the global annual mean surface temperature and net top of atmosphere radiative flux timeseries in response to an assumed $F_{4x} = 7.2 \ Wm^{-2}$ step change in forcing ($F_{4x}$ corresponding approximately to a quadrupling of $CO_2$, Zhang and Huang 2014), $\tau_n$ is the decay time associated with the timescale $n$, $S_n$ and $R_n$ are scaling factors and $T_0$ and $R_{extrap}^{4x}$ are constant terms. $T_0$ represents the pre-pulse temperature, taken here as the mean temperature in the last available 500 years of the control simulation. $R_{extrap}^{4x}$ is the radiative flux imbalance as $t \to \infty$ in the forced simulation and is calibrated during the calculation.

We distinguish between the radiative flux imbalance in the PICTRL ($R^{CTRL_0}$) and in the asymptotic limit of the ABRUPT4X simulations ($R_{extrap}^{4x}$). For models which provided constant forcing extensions of transient experiments, we assume $R_{extrap}^{4x}$ is a fixed property of the fitted pulse-response function. $R^{CTRL_0}$ is calculated as the time average of net Top of Atmosphere (TOA) flux from the last 500 years of *PICTRL*. In fully equilibrated models with no energetic leaks, it would be expected that $R_0^{CTRL} = 0$, but it has been noted previously that this is not always the case and small energetic imbalances remain in some models even after the model global mean temperature trends have ceased (Rugenstein et al., 2019).

Existing studies differ in the number of independent equilibration timescales ($N$) which describe the joint evolution of top of atmosphere net radiative balance ($R_p(t)$) and the global mean surface temperature ($T_p(t)$) in response to a step change in forcing, generally using 2 (Smith et al., 2018; Rugenstein and Armour, 2021) or 3 timescales (Proistosescu and Huybers, 2017; Rugenstein and Armour, 2021; Caldeira and Myhrvold, 2013) timescales. Here we consider solutions ranging from 2 to 5 timescales allowing for a range of thermal responses corresponding approximately to sub-decadal, decadal, centennial, millennial and multi-millennial (see Tables 1 and 2).

**Table 1.** Table showing the included modes from Table 2 for each model variant considered

| Model | sub-decadal | decadal | centennial | millennial | multi-millennial |
|---|---|---|---|---|---|
| **2 timescale** | ✓ | ✓ | | | |
| **3 timescale** | ✓ | ✓ | ✓ | | |
| **4 timescale** | ✓ | ✓ | ✓ | ✓ | |
| **5 timescale** | ✓ | ✓ | ✓ | ✓ | ✓ |

For LongrunMIP models which provide an experiment with an abrupt quadrupling of $CO_2$ (*ABRUPT4X* hereon), we take $T_p(t)$ and $R_p(t)$ as global annual mean values from *ABRUPT4X* simulations to directly calibrate the parameters in Equations 1a and 1b. Some models, however, do not provide *ABRUPT4X*, instead providing constant forcing extensions of other climate change experiments (see Rugenstein et al. 2020). For these models, we further assume a linear pulse-response formulation to represent the thermal global mean response to the corresponding forcing time-series as the convolution of the thermal response to a step change in forcing, combined with the forcing timeseries itself (Joos et al., 2013).

$$T(t) = \sum_{t'=1}^{t} T_p(t - t') \frac{F(t') - F(t' - 1)}{F_{4x}} \tag{2a}$$

$$R(t) = \sum_{t'=1}^{t} R_p(t - t') \frac{F(t') - F(t' - 1)}{F_{4x}} \tag{2b}$$

where $F(t)$ is the forcing time series of the corresponding experiment. Here we assume approximate logarithmic forcing dependencies (Myhre et al., 1998) for carbon dioxide (a dependency which is an empirical outcome of more complex radiative transfer models; Huang and Bani Shahabadi 2014) and integrated forcing estimates (Meinshausen et al., 2011) for the one model (ECEARTH) which extended a multi-forcer future scenario experiment in LongRunMIP. The latter forcing estimate is an approximation with central estimates for aerosol and greenhouse gas forcing rather than model-specific values, but the effective forcing timeseries experienced by ECEARTH under RCP85 is not knowable without dedicated simulations (Pincus et al., 2016).

## 2.1 Bayesian Calibration of model response parameters

We fit the response equations detailed in Eqs. 2a and 2b to the output of each ensemble member's global mean radiative flux and surface temperature timeseries using a Markov Chain Monte Carlo optimizer (Foreman-Mackey et al. 2013; as implemented in the 'lmfit' Python module), sampling models which allow for a range of $N = [2, 3, 4, 5]$ representative decay timescales.

**Table 2.** Parameters and prior ranges considered in the Bayesian calibration of Eq. 1a. Parameters marked * are optionally included according the model under consideration (see Table 1)

| Parameter | long name | Units | Min value | Max value |
|---|---|---|---|---|
| $S_1$ | Subdecadal timescale sensitivity | K | 0 | 10 |
| $S_2$ | Decadal timescale sensitivity | K | 0 | 10 |
| $S_3*$ | Centennial timescale sensitivity | K | 0 | 10 |
| $S_4*$ | Millennial timescale sensitivity | K | 0 | 10 |
| $S_5*$ | Multi-millennial timescale sensitivity | K | 0 | 10 |
| $R_1$ | Subdecadal timescale energetic scaling | $Wm^{-2}$ | -10 | 10 |
| $R_2$ | Decadal timescale energetic scaling | $Wm^{-2}$ | -10 | 10 |
| $R_3*$ | Centennial energetic scaling | $Wm^{-2}$ | -10 | 10 |
| $R_4*$ | Millennial energetic scaling | $Wm^{-2}$ | -10 | 10 |
| $R_5*$ | Multi-millennial energetic scaling | $Wm^{-2}$ | -10 | 10 |
| $\tau_1$ | Subdecadal timescale | years | 0 | 10 |
| $\tau_2$ | Decadal timescale | years | 10 | 100 |
| $\tau_3*$ | Centennial timescale | years | 100 | 1000 |
| $\tau_4*$ | Millennial timescale | years | 1000 | 5000 |
| $\tau_5*$ | Multi-millennial timescale | years | 5000 | 100000 |
| $R_{extrap}^{4x}$ | Asymptotic energy imbalance | $Wm^{-2}$ | -10 | 10 |

**Table 3.** Table showing assumed forcing evolution for experiments in LongRunMIP. (*) Logarithmic $CO_2$ forcing dependency is assumed following Myhre et al. (1998). (**) $F_{historical}(t)$, $F_{RCP85}(t)$ forcing is taken according to Meinshausen et al. (2011).

| Scenario | $F(t)$ ($Wm^{-2}$) | Time range (years) | Models | Forcing scaling for $\Delta T_{best-est}$ |
|---|---|---|---|---|
| **ABRUPT4X** | $0$ <br> $(5.35)log(4)$ | $(t < 0)$ <br> $(t \geq 0)$ | CCSM3, CESM104, CNRMCM61, ECHAM5MPIOM GISSE2R, HadCM3L, HadGEM2, IPSLCM5A, MPIESM11, MPIESM12 | 1 |
| **1pct2x*** | $0$ <br> $(5.35)log(1.01^t)$ <br> $(5.35)log(2)$ | $(t < 0)$ <br> $(0 \leq t < 70)$ <br> $(t \geq 70)$ | GFDLCM3, GFDLESM2M | 2 |
| **1pct4x*** | $0$ <br> $(5.35)log(1.01^t)$ <br> $(5.35)log(4)$ | $(t < 0)$ <br> $(0 \leq t < 140)$ <br> $(t \geq 140)$ | MIROC32 | 1 |
| **RCP85**** | $F_{historical}(t)$ <br> $F_{RCP85}(t)$ <br> $F_{RCP85}(2300)$ | $(t < 2005)$ <br> $(2005 \leq t < 2300)$ <br> $(t \geq 2300)$ | ECEARTH | 0.583 |

## 3  Results

### 3.1  Assessment of model response timescale

The following section is used to assess the simplest acceptable multi-timescale model for the emulation of different ESMs in the LongRunMIP archive. We quantify this using the Root Mean Square Error (RMSE) associated with the least-square fit optimization (assessed as the best performing member of the MCMC posterior solution). If the addition of an additional, longer timescale in the fit corresponds to a reduction in combined RMSE of 0.5% or more - the longer timescale model is used.

The performance of fitted multi-timescale models for $GMT$ (Global Annual Mean Surface Temperature) and $NET$ (Global Annual mean net Top of Atmosphere radiative imbalance) timeseries is summarised in Figure 1, which shows the combined error in the fits for $GMT$ and $NET$ associated with the absolute least-square fit for each of the model variants described in Table 1. The associated timeseries for the best fitted model in the context of the original model data for $GMT$ and $NET$ are shown in supplemental Figures (Figures A1 and A2).

We find that for all LongRunMIP models, the N=2 timescale model performs significantly worse than N≥3 timescale models allowing for centennial and longer response timescales. This is both evident by the significantly larger best fit errors (Figure 1) as well as visibly poor fits (Supplemental Figures A1 and A2).

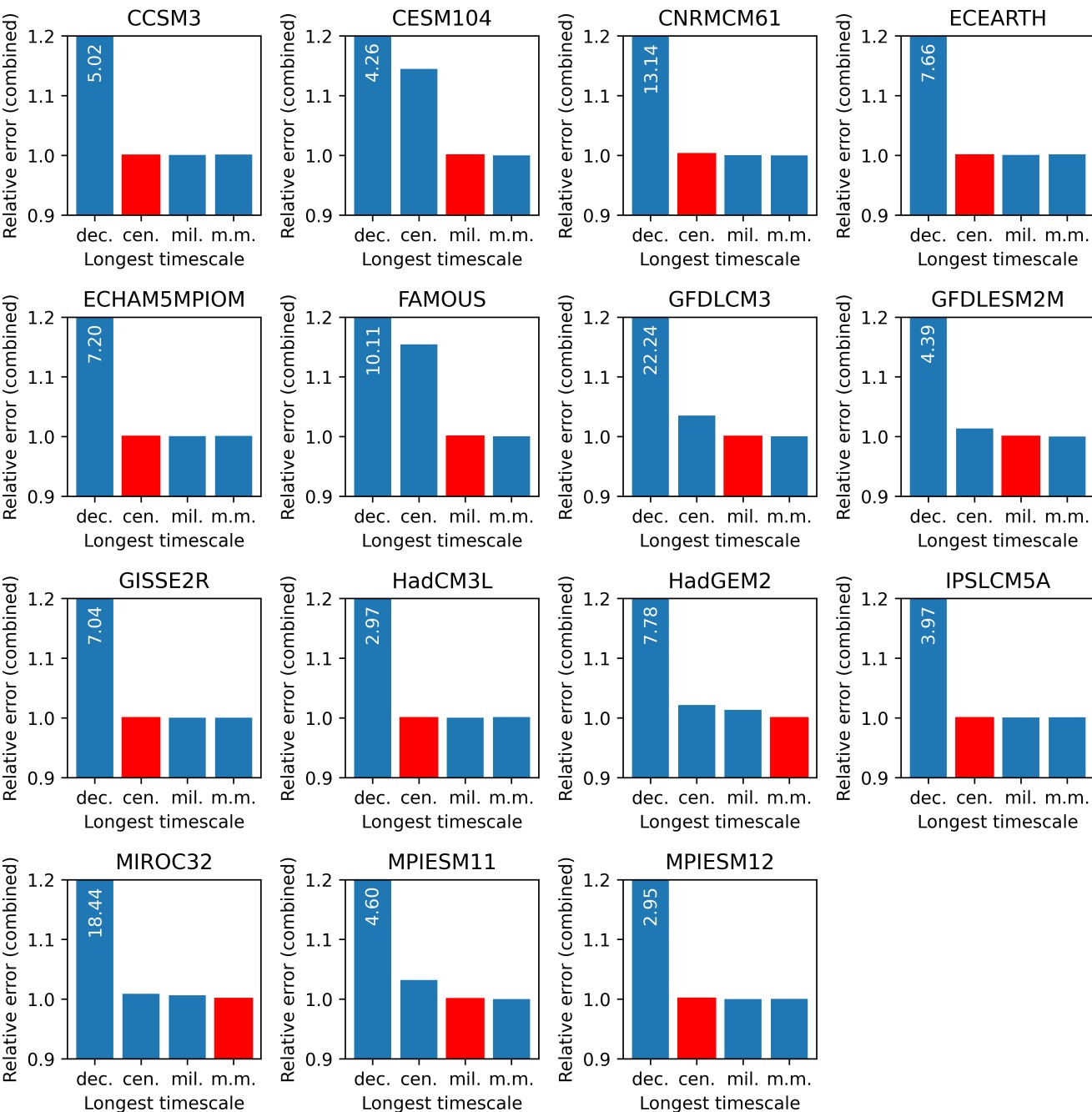

**Figure 1.** Illustration of the Root Mean Square Error for the fit to global mean temperature and net TOA radiative balance using models allowing for a range of timescales. Dec., Cen., Mil., and m.m. are Decadal, Centennial, Millenial, and Multi-millenial timescale models respectively. RMSE values for each variable (NET and GMT) are normalised relative to the best overall fit for that variable, each multiplied by 0.5 to give a combined error. The shortest timescale model with errors within 0.5 percent tolerance of the overall best performing model is illustrated in red. Included modes and parameter priors are detailed in Table 1 and 2). In cases where error is truncated by the vertical axis, value is printed in white.

Differences between the N=3, 4 and 5 timescale models are dependent on the ESM being fitted. For some models (CCSM3, CNRMCM61, ECEARTH, ECHAM5MPIOM, GISSE2R, HadCM3L, IPSLCM5A, MPIESM12), no significant improvement in fit is seen beyond the centennial timescale model (Figure 1). For other models, fits are further improved by allowing a millennial (CESM104,FAMOUS, GFDLCM3) or multi-millennial timescale (HadGEM2, MIROC32). Parameters associated
with the best fitting models are listed in Supplemental Table A1, and fitted MCMC ensembles corresponding the selected class of model illustrated in red in Figure 1 are carried through for the remainder of the study.

### 3.2   Assessment of climate sensitivity

The conventional effective climate sensitivity (*EffCS*) is calculated using the first 150 years of simulation, linearly extrapolating GMT as a function of NET to $R_0^{CTRL}$. Control global mean temperatures and TOA energetic imbalances are expressed as
anomalies relative to $T_0$. We assess errors *EffCS* due to state-dependent radiative imbalance by calculating $EffCS_{corr}$, where feedbacks in the first 150 years are instead linearly extrapolated to $R_{4x}^{extrap}$.

A third estimate of equilibrium warming, $\Delta T_{best-est}$, follows Rugenstein et al. (2020), by calculating the effective climate sensitivity based on the years corresponding to the last 15% of warming in the simulation (that is, for all years following the point when the simulation first exceeds 85% of the average global mean temperature anomaly in the last 20 years of the
*ABRUPT4X* simulation). For models which do not directly provide *ABRUPT4X* (GFDLCM3, GFDLESM2M and MIROC32), $\Delta T_{best-est}$ is calculated by scaling by the ratio of radiative forcing in *ABRUPT4X* relative to that in the multi-thousand year constant forcing period in the experiment provided (following Rugenstein et al. (2020), see Table 3).

We finally calculate a fourth estimate of climate sensitivity $\Delta T_{extrap}$ as in Eq. 3 in the equilibrated (*ABRUPT4X*) simulation using the ensemble of fitted parameters from Bayesian calibration of Equation 1a, using again global mean temperature
anomalies from *ABRUPT4X* relative to $T_0$ (taken as mean temperatures over the last 100 years of *PICTRL*).

$$T_{extrap} = \sum_{n=1}^{N} S_n + T_0 \tag{3}$$

We estimate the long term radiative imbalance in the *ABRUPT4X* simulation from the fitted values for $R_{extrap}^{4x}$ (along with $R_n$, the amplitude of the decay in forcing at the timescale corresponding to $\tau_n$) from Eq. 1b. Previous studies have assumed in the calculation of $\Delta T_{best-est}$ that $R_{extrap}^{4x} = R_0^{CTRL}$ (Rugenstein et al., 2020), an assumption we test here.
We follow convention by reporting climate sensitivities for a doubling of carbon dioxide from pre-industrial levels. As such, we follow standard practice in dividing ABRUPT4X sensitivities by 2 to obtain (EffCS, $\Delta T_{extrap}$ and $\Delta T_{best-est}$) (Meehl et al., 2020), though we note that in some models this approximation introduces minor errors (Jonko et al. (2012); Bloch-Johnson et al. (2021), these are not the focus of the present study.

### 3.3   Relevance of energetic leakages

We consider first the radiative tendencies of the models in the climate change experiments, compared with the control state. Figure 2 shows the evolution of the top of atmosphere net radiative imbalance in the LongRunMIP climate change experiments, as well as the control simulation - together with the projected evolution of a simulated ABRUPT4X simulation using the fitted multi-timescale model. We note that there is significant model diversity in the behaviour of models in the approach to equilibrium. Some models (CESM104, GISSE2R, GFDLESM2M, GFDLCM3 and MPIESM11) behave as expected, showing
$R_0^{CTRL} = 0$ and $R_{4x}^{extrap} = 0$ (Figure 2).

A second class of model exhibits a radiative imbalance in the control simulation, but the *ABRUPT4X* simulation converges to the same state ($R_0^{CTRL} = R_{4x}^{extrap} \neq 0$ e.g. MIROC32, MPIESM11). Finally, a third class appears to converge to different states in *PICTRL* and *ABRUPT4X* ($R_0^{CTRL} \neq R_{4x}^{extrap}$ e.g. CCSM3, CNRMCM61, ECEARTH, HadCM3L, MIROC32, MPIESM12 and IPSLCM5A) - implying that Effective Climate Sensitivity may be biased in these models if calculated assuming that the
ABRUPT4X simulation is tending towards the equilibrium radiative state of the PICTRL simulation.

Figures 3 and 4 show the impact on these biases on the derived value for Equilibrium Climate Sensitivity. The relationship between temperature and TOA fluxes for the fitted multi-timescale models for *ABRUPT4X* simulations in the LongRunMIP archive are presented in Figure 3, while Figure 4 shows the temperature evolution as a function of time.

Models with exact agreement between $R_0^{CTRL}$ and $R_{4x}^{extrap}$ also tend to exhibit similar values for $\Delta T_{best-est}$ and $\Delta T_{extrap}$
and in cases where there is little or no difference in feedbacks in the early and late stages of the simulation (e.g. CESM104, GISSE2R, MPIESM11), *EffCS* is also similar to $\Delta T_{best-est}$ and $\Delta T_{extrap}$. Other models (e.g. ECEARTH, ECHAM5MPIOM, FAMOUS, GFDLCM3, GFDLESM2M, IPSLCM5A) show significant differences in early and late stage feedbacks, manifested as a $\Delta T_{best-est}$ which differs from *EffCS*.

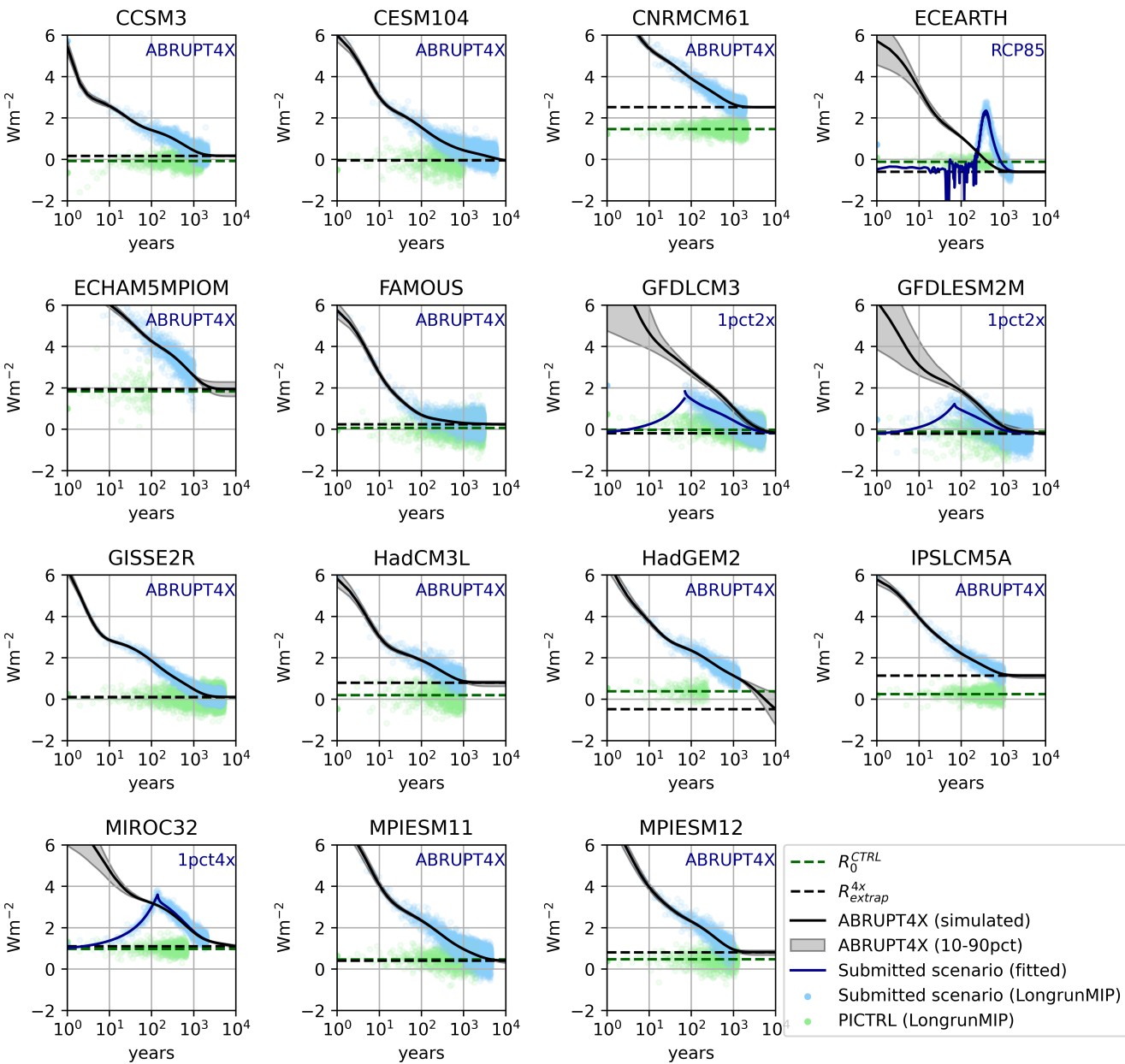

**Figure 2.** Top of atmosphere net radiative imbalance plotted as a function of time (log scale) for the members of the LongrunMIP ensemble. Dashed green line shows the control radiative imbalance ($R_0^{CTRL}$), while dashed black line shows the predicted ABRUPT4X radiative imbalance ($R_{extrap}^{4x}$). Semi-transparent blue and green points show annual mean upgoing net radiative flux from *PICTRL* and the submitted simulation (printed in blue text) respectively. Black line shows the simulated response to ABRUPT4X for the multi-timescale model, while shaded grey regions and thin lines show the 10th and 90th percentiles of the fitted ensemble projections for ABRUPT4X. If the submitted simulation was not ABRUPT4X, the thick blue line shows the MCMC posterior median TOA timeseries for the submitted simulation using the chosen multi-timescale model (see Table 1)

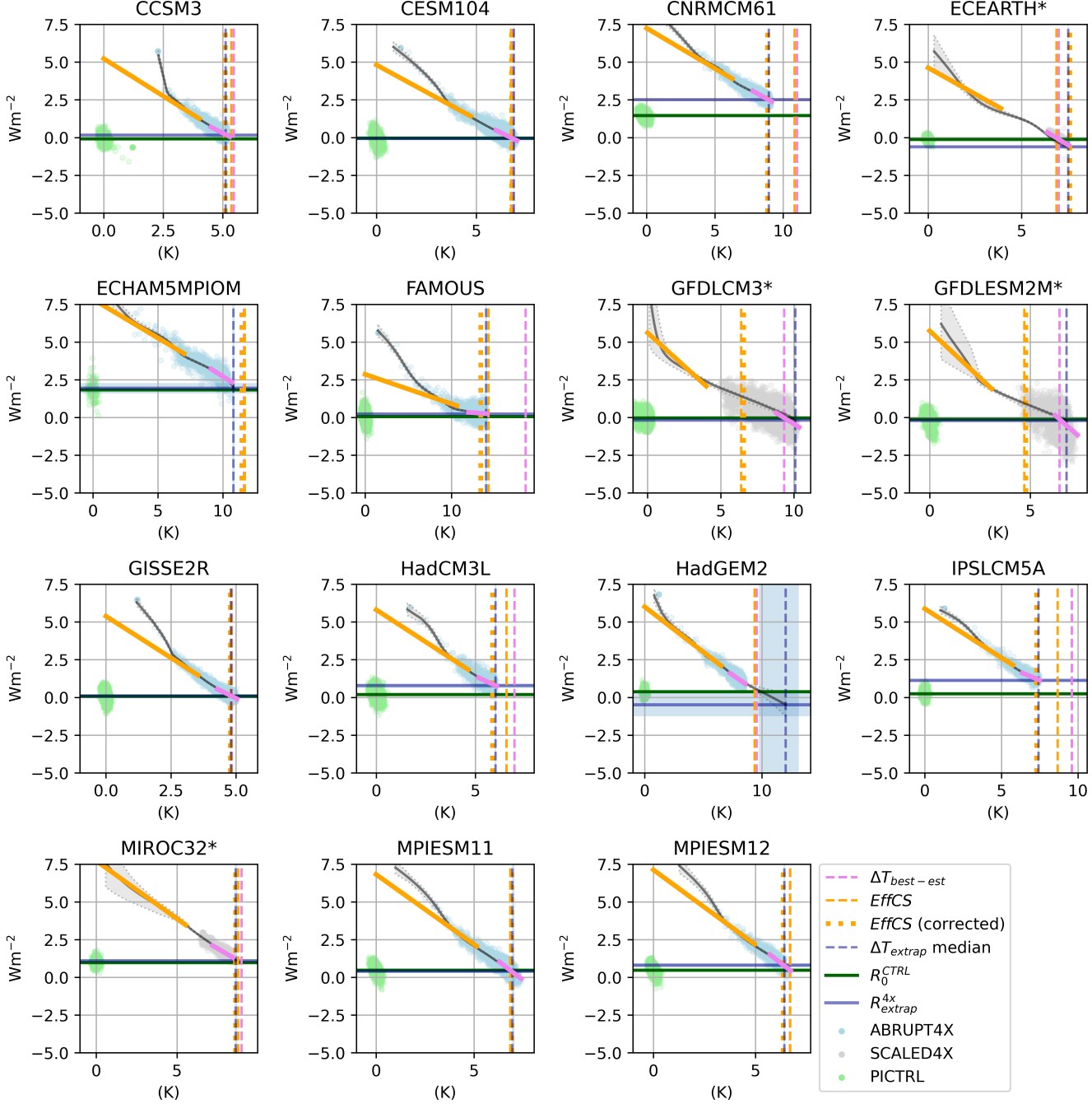

**Figure 3.** Global mean net radiative imbalance as a function of surface temperature for different members of the LongrunMIP archive. Vertical axis shows absolute top of atmosphere net radiative imbalance, horizontal axis shows surface temperature relative to the final 500 years of the control simulation. Models marked '*' did not provide ABRUPT4X directly (see Table 3). Solid black lines show the median simulation of ABRUPT4X for the fitted MCMC posterior of the multi-timescale model, shaded grey areas show 5-95% confidence intervals. Light blue points are individual years from *ABRUPT4X* (if available). For * models, grey points show years in the latter portion of the simulation after which forcing is constant, scaled according to Table 3. Light green points are annual means from *PICTRL*. Yellow solid line shows the regression fit in years 0-150 for the original ABRUPT4X data if available (or simulated ABRUPT4X median model for models marked '*'), corresponding to the *EffCS* dashed yellow vertical line and *EffCS* (corrected) dotted yellow vertical line. Purple solid line shows regression fit to the last 15% of warming following Rugenstein et al. (2020), corresponding to the $\Delta T_{best-est}$ vertical dashed line. Horizontal green line shows *PICTRL* net energy imbalance averaged over the final 500 years of the simulation. Horizontal solid blue line shows $R_{extrap}^{4x}$ while vertical dashed blue line shows $\Delta T_{extrap}$, shaded areas illustrate uncertainty in these values.

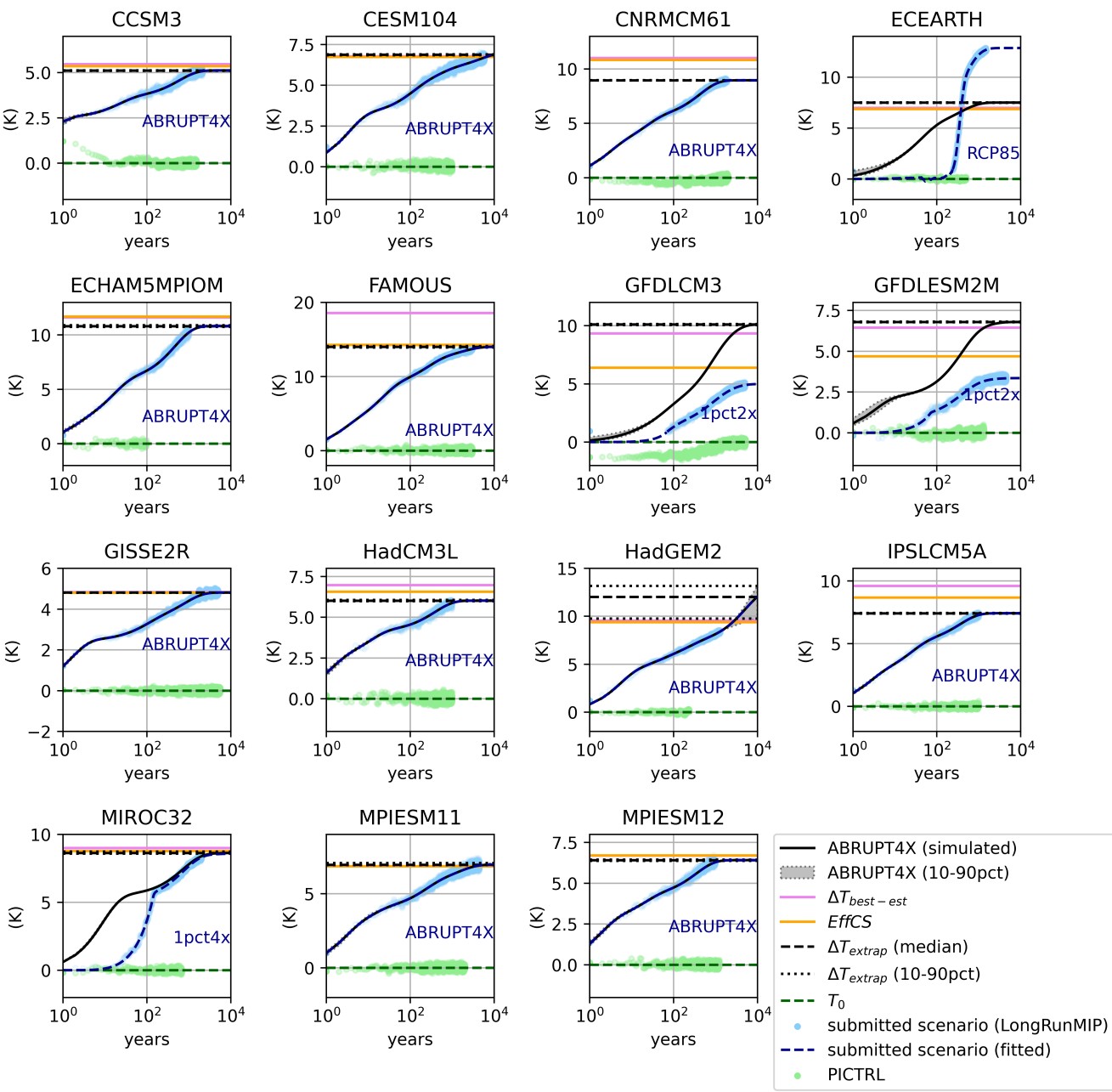

**Figure 4.** Global mean temperature anomaly with respect to the last 500 available years of the *PICTRL* simulation, plotted as a function of time (log scale) for the members of the LongrunMIP ensemble. Green points show annual global mean surface temperature anomalies from the LongRunMIP *PICTRL* simulation, while blue points show data from the submitted climate change experiment (printed in blue text for each model). Thick blue lines show the median top of atmosphere timeseries using the MCMC posterior fit for the multi-timescale model selected to represent the corresponding ESM (see Section 2 and Figure 1). Black lines show the median response of the fitted multi-timescale model to an ABRUPT4X forcing, while shaded grey regions and thin dotted lines show the 10th and 90th percentiles of the fitted ABRUPT4X ensemble projections. Dashed black horizontal line illustrates $\Delta T_{extrap}$ (median), yellow solid is *EffCS*, pink solid is $\Delta T_{best-est}$, dashed green shows $T_0$. Readers should note y-axis differs by subplot.

| Model | Years | *EffCS* | *EffCS*$_{corr}$ | Difference | $\Delta T_{best-est}$ | $\Delta T_{extrap}$ | $R^{4x}_{extrap}$ | $R^{CTRL}_0$ |
|---|---|---|---|---|---|---|---|---|
| CCSM3 | 2120 | 2.68 | 2.55 | -0.13 | 2.73 | 2.56 (2.55,2.56) | 0.09 (0.07,0.1) | -0.04 |
| CESM104 | 5900 | 3.37 | 3.38 | 0.01 | 3.39 | 3.44 (3.43,3.45) | -0.03 (-0.04,-0.01) | -0.02 |
| CNRMCM61 | 1850 | 5.42 | 4.42 | -0.99 | 5.51 | 4.47 (4.47,4.48) | 1.26 (1.25,1.27) | 0.73 |
| ECEARTH | 1271 | 3.44 | 3.79 | 0.35 | 3.50 | 3.75 (3.74,3.76) | -0.3 (-0.32,-0.28) | -0.06 |
| ECHAM5MPIOM | 1001 | 5.84 | 5.73 | -0.10 | 5.81 | 5.4 (5.37,5.45) | 0.97 (0.79,1.14) | 0.92 |
| FAMOUS | 3000 | 7.13 | 6.68 | -0.45 | 9.27 | 6.99 (6.96,7.05) | 0.12 (0.09,0.13) | 0.03 |
| GFDLCM3 | 5000 | 3.19 | 3.28 | 0.08 | 4.66 | 5.05 (5.02,5.07) | -0.09 (-0.11,-0.05) | -0.01 |
| GFDLESM2M | 4500 | 2.35 | 2.39 | 0.04 | 3.22 | 3.4 (3.39,3.41) | -0.1 (-0.13,-0.09) | -0.05 |
| GISSE2R | 5001 | 2.40 | 2.39 | -0.01 | 2.42 | 2.4 (2.4,2.41) | 0.05 (0.05,0.06) | 0.04 |
| HadCM3L | 1000 | 3.28 | 2.93 | -0.35 | 3.48 | 3.01 (2.99,3.03) | 0.4 (0.31,0.43) | 0.10 |
| HadGEM2 | 1299 | 4.69 | - | - | 4.77 | 6.01 (4.87,6.58) | -0.24 (-0.63,0.16) | 0.19 |
| IPSLCM5A | 1000 | 4.33 | 3.65 | -0.68 | 4.80 | 3.71 (3.69,3.72) | 0.57 (0.51,0.59) | 0.12 |
| MIROC32 | 2002 | 4.39 | 4.31 | -0.08 | 4.49 | 4.31 (4.29,4.36) | 0.55 (0.52,0.58) | 0.49 |
| MPIESM11 | 4459 | 3.42 | 3.46 | 0.03 | 3.42 | 3.47 (3.46,3.53) | 0.21 (0.14,0.23) | 0.24 |
| MPIESM12 | 1000 | 3.35 | 3.18 | -0.17 | 3.34 | 3.21 (3.19,3.22) | 0.41 (0.34,0.45) | 0.24 |

**Table 4.** Fitted parameters and uncertainties for the LongrunMIP experiments. Median values, with 5th and 95th percentiles in brackets where relevant. The Difference column shows *EffCS*$_{corr}$-*EffCS*. *EffCS*$_{corr}$ is not calculated for HadGEM2 due to large uncertainties in $R^{4x}_{extrap}$

Models with significant differences between $R^{CTRL}_0$ and $R^{extrap}_{4x}$ (CNRMCM61, FAMOUS, ECEARTH, HadCM3L, IP-SLCM5A, MPIESM12), exhibit similar biases in both $\Delta T_{best-est}$ and *EffCS*. For example, CNRMCM61 exhibits relatively constant feedbacks on century and millennial timescales, so $\Delta T_{best-est}$ and *EffCS* are similar (5.42K, 5.51K respectively), but $\Delta T_{extrap}$, which is well fitted by the data is significantly lower (4.47±0.01K) (Figure 4 and Table 4) due to the differing esti-
5 mated equilibrium energetic imbalance in *ABRUPT4X* and *PICTRL* simulations. The fitting process for HadGEM2 determined that a multi-millennial response mode was necessary, which remains unconstrained by the fit so it is not possible to estimate $\Delta T_{extrap}$ with confidence for this model (the simulation length for HadGEM2 is 1299 years, so it remains possible that a 5000 year simulation as provided by a number of other models could rule out the need for the multi-millennial response mode).

Of these models with apparently state-dependent energetic balance, some (HadCM3L, FAMOUS, ECEARTH) appear to
10 show a control simulation where $R^{CTRL}_0 \approx 0$, but an *ABRUPT4X* simulation which converges to a state of energetic imbalance (Figure 2). This, in turn introduces a source of potential bias in the estimate of effective climate sensitivity if the system is converging to a non-equilibrated state, implying that the control simulation may be tuned to exhibit energetic balance but the equilibrated 4xCO$_2$ state is subject to an energy leak. A particularly extreme example is FAMOUS, where a small difference in extrapolated energetic balance, combined with large feedback parameter results in a much larger values of $\Delta T_{best-est}$ (9.27K)
than $\Delta T_{extrap}$ (6.99K, see Table 4, Figure 4) or *EffCS* (7.13K) [1]. Similarly for HadCM3L, the fitted extrapolated sensitivity $\Delta T_{extrap}$ (3.03K, see Table 4 and Figure 4) is lower than $\Delta T_{best-est}$ (3.49K) and *EffCS* (3.29K).

$\Delta T_{extrap}$ differs from *EffCS* both due to the presence of state dependent energetic biases, but also due to feedbacks which occur over the multi-thousand year timescales resolved in the LongrunMIP experiments. We can isolate the bias in *EffCS* induced by state-dependent energetic imbalance in the LongrunMIP cases by using a different extrapolated energetic state
(Figures 5, 3). As in the standard calculation of *EffCS*, we take a least-squares linear fit of temperature as a function of $N$ in the first 150 years, but instead linearly extrapolating to $N = R^{4x}_{extrap}$ rather than $N = R^{CTRL}_0$ in the standard calculation to produce a bias corrected *EffCS*$_{corr}$. We find that 2 models in LongRunMIP are significantly impacted by this correction (see Figure 5 and Supplemental Figure A3) - CNRMCM61 ($EffCS = 5.42K$, $EffCS_{corr} = 4.42K$) and IPSLCM5A ($EffCS = 4.33K$, $EffCS_{corr} = 3.65K$). A number of other models are impacted to a lesser extent (see Table 4).
The analysis was repeated for the wider CMIP5 and CMIP6 ensembles. However, the standard CMIP5 and CMIP6 simulations are insufficiently long to fit response timescales of centennial or longer - hence $\Delta T_{extrap}$ (or $R^{4x}_{extrap}$) are not constrained using the multi-timescale fitting approach (see Figure 5). It is notable that flux imbalances are present in the control state of a number of models in both CMIP5 and CMIP6, but longer simulations are required to assess if these represent structural imbalances or an insufficiently long spinup. The centennial and longer timescales are not constrained in 150 year simulations, hence
it is not possible to estimate $\Delta T_{extrap}$ and $R^{4x}_{extrap}$ with any confidence. We note, however, that in most cases the uncertainties in the fitted 3 timescale solution generally allow for equilibrium values which are higher then the effective climate sensitivity

---

[1]Using $R^{4x}_{extrap} = -0.16 W m^{-2}$ rather than $R^{CTRL}_0 = -.01 W m^{-2}$ would result in a value of $\Delta T_{best-est} = 7.01K$, broadly consistent with *EffCS* and $\Delta T_{extrap}$

as assessed over the first 150 years of simulation. Only a small number of models allow for fitted solutions which have a lower $\Delta T_{extrap}$ than the *EffCS* (CESM2, CCSM4, MIROC5, CNRMESM2.1, ACCESS-CM2). One of these cases (CNRMCM6.1) is a close relative of the CNRMESM2.1 - the LongrunMIP simulation which we identified to be potentially subject to biases owing to energetic imbalances in the $4xCO_2$ equilibrium state.

## 4   Conclusions

We have considered an alternative approach for calculating long term tendencies of temperature and planetary energetic imbalance from simulations in which atmospheric carbon dioxide concentrations are instantaneously perturbed. This approach relies on the assumption that the evolution of the system can be represented as a sum of decaying exponential terms with differing timescales. An existing project, LongrunMIP, provides multi-millennial simulations which allow for the fitting a multi-timescale simple model which allows for annual, decadal, centennial and millennial responses.

We find that this approach highlights some potential limitations and biases associated with using effective climate sensitivity to predict equilibrium warming. It has been observed before that energetic imbalances exist in some models in the CMIP archive (Rugenstein et al., 2019; Hobbs et al., 2016; Irving et al., 2021), and in this study we show that such control state radiative imbalances are relatively widespread in CMIP5 and CMIP6. The conventional assumption used to calculate effective climate sensitivity in these cases is that such imbalances remain constant, such that radiative anomalies from the control state can be used to calculate the effective climate sensitivity. Critically, in some LongrunMIP simulations, we observe that energetic imbalances are themselves state-dependent. This undermines the concept of effective climate sensitivity - if we do not know what the radiative imbalance will be when temperatures stabilise in an *ABRUPT4X* simulation, we in turn cannot predict the climate sensitivity (using this method) with precision.

In practice, only some models in CMIP5 and CMIP6 appear to exhibit significant radiative imbalances in the control state (see Figure 5), and although the 150 year *ABRUPT4X* simulations are insufficient to assess if these energetic imbalances are state-dependent, these are the cases where we might be least confident in the effective climate sensitivity value. Models may exhibit non-equilibrium fluxes in the control state for a number of different reasons - either the model has not been run for sufficiently long in the control configuration to reach a state of energetic balance, or there is a persistent energetic leak in the model, which may be constant or evolving (Hobbs et al., 2016). In either case, the results presented in this study draw into doubt whether such imbalances can be assumed to remain constant in a climate perturbed through alteration of climate forcers.

Further, we find that some models which are in or close to energetic balance in the control state do not converge to energetic balance following the step change in climate forcing. This implies that models fall into two potential categories: those where the energetic budget of the model is structurally closed through the elimination of all leaks, and those where the model parameters have been adjusted to produce near-zero net TOA fluxes in the control state. The latter case is still potentially subject to errors in the estimation of effective climate sensitivity, because if energetic imbalances are dependent on climate forcers, then the calibrated minimisation of net TOA fluxes may be inappropriate for the perturbed climate state. A simple analysis of the net fluxes in the control simulation cannot distinguish between structurally balanced models and tuned balanced models - but centers which operationally adjust parameters to minimize energetic losses should be aware of this potential bias in effective climate sensitivity.

Models with state dependent energetic imbalance will not reach true energetic equilibrium (as defined by a state of radiative balance of the system) in response to a climate forcing . This still allows for the model to reach an asymptotic stable state (effectively including an energy leak) but it does not allow for the derivation of effective climate sensitivity which requires prior knowledge of the asymptotic equilibrium TOA balance. The method suggested here presents an alternative approach for deriving climate sensitivity, but it is clearly less than ideal - requiring simulations of 5000 years of simulation to produce a stable estimate for some models. We must also consider the possibility for these models that there is no stable state. If energy leaks are a function of the climate state, and the system is not tending towards a state of radiative equilibrium, our evidence that models are converging to a stable temperature is empirical and longer simulations will be required to investigate these multi-millennial dynamics and confirm that a stable asymptotic solution exists.

Our results highlight the potential for error in the estimation of effective climate sensitivity through the assumptions on the asymptotic radiative balance of climate models. In the case of LongrunMIP - there is a significant difference between the distribution of fitted asymptotic values of energetic imbalance in *ABRUPT4X* compared with the mean energetic balance in *PICTRL* in 11 of 15 models (see Table 4). In 5 out of 15 cases, this results in a bias in Effective Climate Sensitivity of 0.3K or more, but this bias is not universally in the same direction. Quantifying the presence of such biases in the wider CMIP6 ensemble is not possible without multi-thousand year control and ABRUPT4X simulations. However, their relatively common occurrence in LongrunMIP suggests that more models could be impacted.

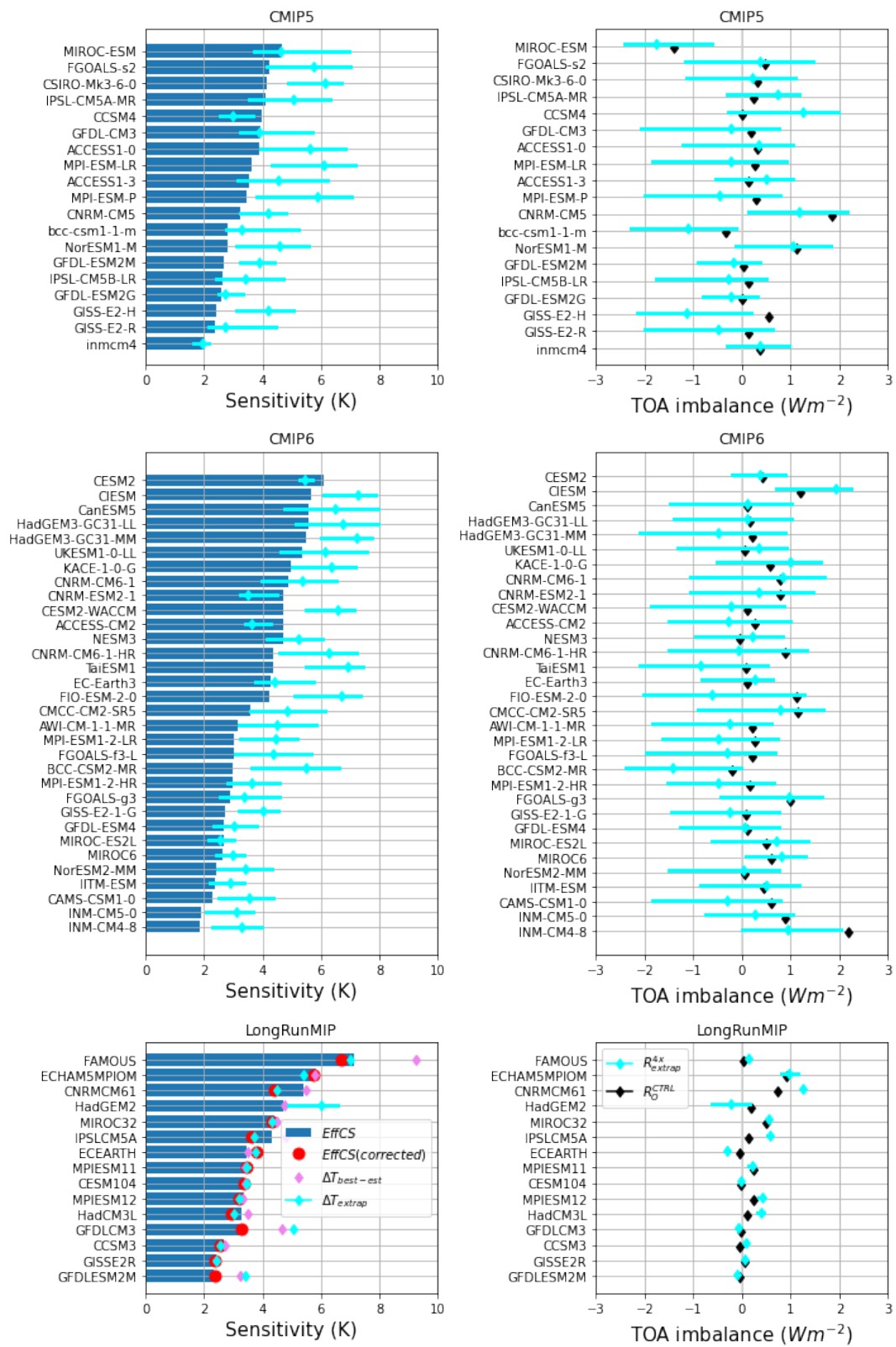

**Figure 5.** Barplots summarising results for three model ensembles, CMIP5 (top row), CMIP6 (middle row) and LongRunMIP (bottom row). Left hand column shows different estimates of equilibrium climate sensitivity. Solid blue bars show EffCS (see text). Light blue diamond and whiskers show the median, 5th and 95th percentiles $\Delta T_{\text{extrap}}$. For LongRunMIP, $\Delta T_{\text{best-est}}$ (following Rugenstein et al. (2020)), is shown in violet diamonds, while $EffCS_{corr}$ is shown with red circles. The right hand column shows $R^{4x}_{\text{extrap}}$ (light blue diamond and whiskers) and $R^{CTRL}_0$ (black diamonds).

This directly impacts our ability to accurately measure *EffCS* from short simulations, and draws into question whether *EffCS* should be used as a factor at all in assessing the fidelity of climate models (Hausfather et al., 2022). Effective climate sensitivity has known limitations that it describes effective feedbacks at a certain representative timescale following a change in forcing (Rugenstein and Armour, 2021), but our results here highlight another issue that *EffCS* can only be used if we can be confident in the asymptotic energetic balance of the model. Such confidence can arise either from a ground-up demonstration of structural energy conservation in the model (Hobbs et al., 2016), or by running sufficiently long simulations to be empirically confident both in the pre-industrial energetic balance and in the asymptotic multi-millennial tendencies of the model following a change in climate forcing. Such experiments are currently difficult to achieve for CMIP class models, the multi-millennial year simulations conducted in Rugenstein et al. (2020) were significantly longer than any experiments conducted previously - and we find in the present study that even a 1300 year simulation is too short to have confidence in the asymptotic state for some models.

Given this, our study has multiple recommendations. Firstly, a greater emphasis in climate model design and quality checking needs to be placed on structural closure of the energy budget in the climate system. Models which can demonstrate that energy is conserved in the model equations can allow confidence that the system as a whole will converge to a state of true radiative equilibrium following a perturbation, which would allow a robust calculation of *EffCS*. For models which cannot demonstrate this, longer simulations are required to be confident in the asymptotic state. These simulations may be prohibitively time and resource consuming. but such limits could potentially be alleviated through the use of lower resolution configurations (Kuhlbrodt et al., 2018; Shields et al., 2012) (with the risk that such models will exhibit different feedbacks from their high resolution counterparts) or by considering analytical approaches to accelerate convergence of complex systems (Xia et al., 2012).

However, in the short term, a more practical approach may be to consider alternative climate metrics which do not require assumptions about the equilibrium state of the system. Transient Climate Response does not require assumptions about radiative flux, but it does not provide direct information on the warming expected under stabilising forcing. A possible alternative is A140 (the warming observed 140 years after a step quadrupling in CO2 concentrations; Sanderson 2020; Gregory et al. 2015), which requires no assumption on equilibrated state - and is more informative on the warming expected under high mitigation scenarios than *EffCS* itself (even if *EffCS* is known without bias due to energetic leaks). In conclusion, the use of Effective Climate Sensitivity as a metric in assessing the response of the climate system should be treated with caution, both due to its lack of relevance to projected warming under mitigation scenarios (Knutti et al., 2017; Frame et al., 2006; Sanderson, 2020) but also due to the fact that its derivation requires assumptions about the asymptotic state of the climate system which do not hold in a number of Earth System Models.

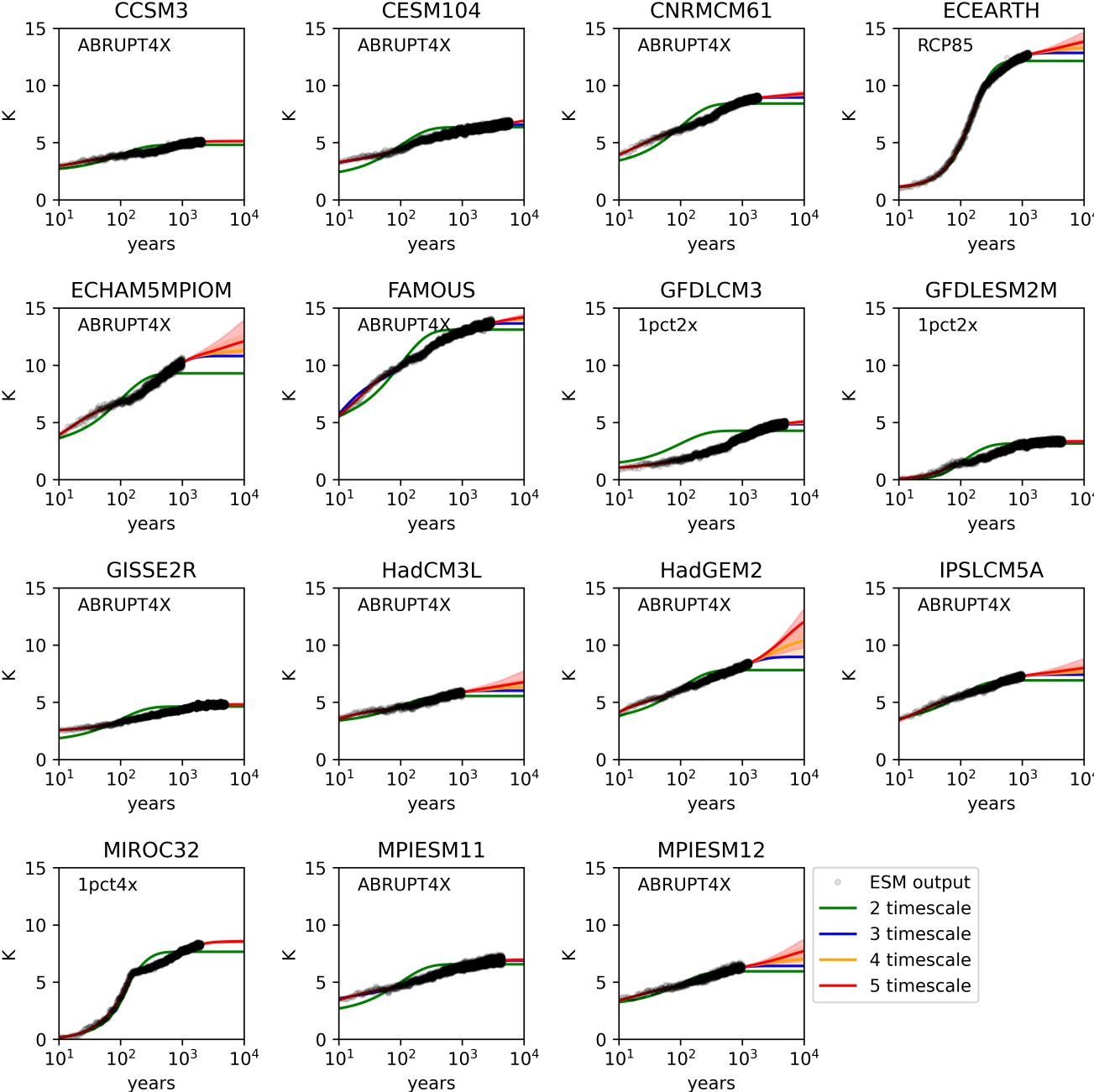

**Figure A1.** Results fitting multi-timescale models to output of LongRunMIP multi-thousand year experiments for global mean surface temperature. Different colors represent different models as detailed in Table 1, shaded areas indicate the 5-95th percentile range in the MCMC fit to the timeseries. Text indicates the model scenario used in the fit (as detailed in table 3).

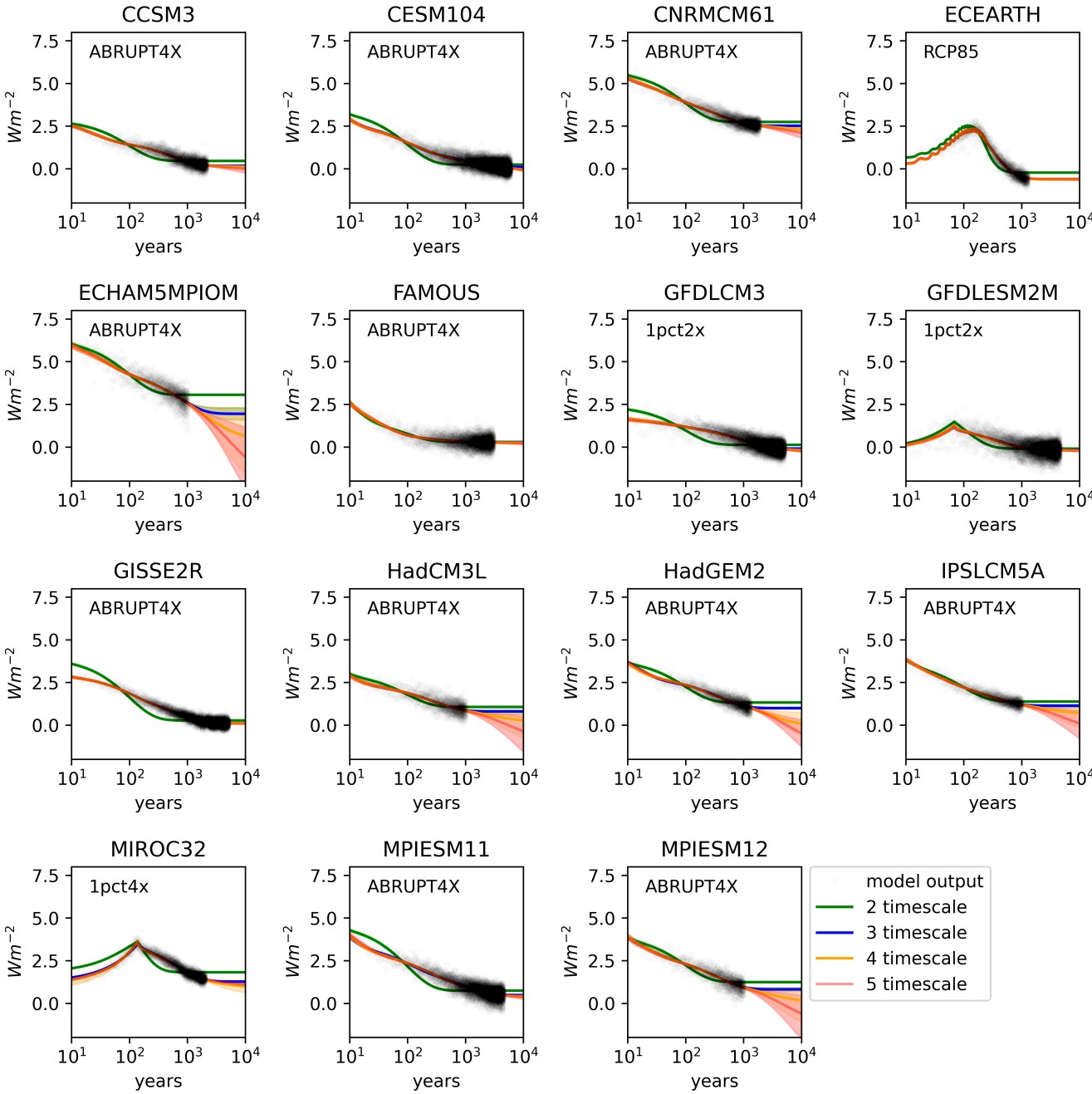

**Figure A2.** Results fitting multi-timescale models to output of LongRunMIP multi-thousand year experiments for global top of atmosphere radiative imbalance. Different colors represent different models as detailed in Table 1, shaded areas indicate the 5-95th percentile range in the MCMC fit to the timeseries. Text indicates the model scenario used in the fit (as detailed in table 3)

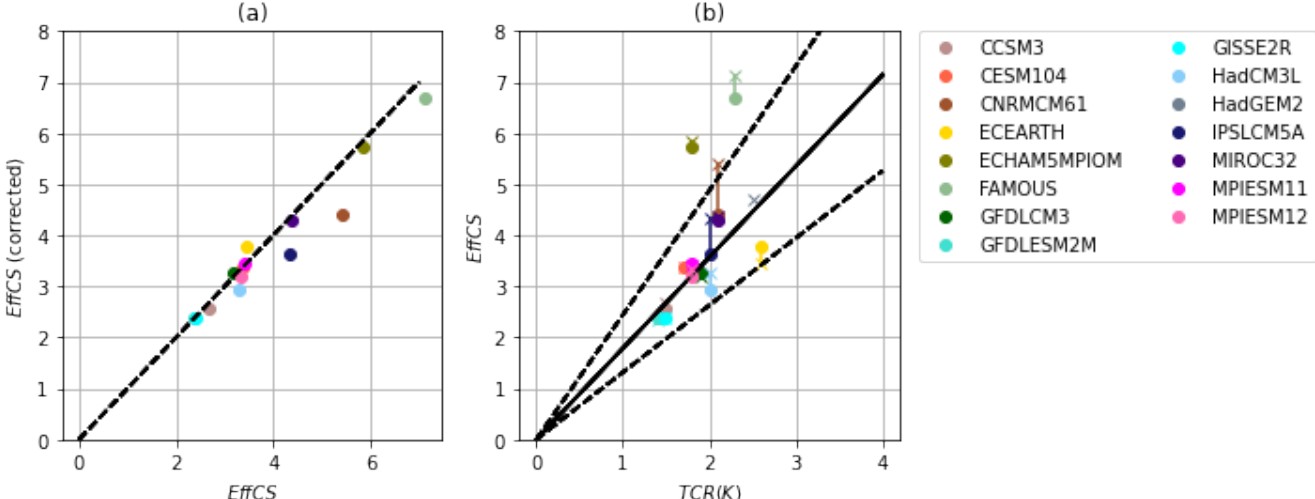

**Figure A3.** (a) Corrected EffCS plotted as a function of uncorrected EffCS. (b) EffCS (crosses) and Corrected EffCS (circles) plotted as a function of Transient Climate Response.

| Model | $S_1$ | $S_2$ | $S_3$ | $S_4$ | $S_5$ | $R_1$ | $R_2$ | $R_3$ | $R_4$ | $R_5$ | $\tau_1$ | $\tau_2$ | $\tau_3$ | $\tau_4$ | $\tau_5$ | $R^{4x}_{extrap}$ |
|---|---|---|---|---|---|---|---|---|---|---|---|---|---|---|---|---|
| CCSM3 | 2.5(2.4,2.6) | 1.1(1.0,1.2) | 1.5(1.5,1.5) | -(-,-) | -(-,-) | 9.0(7.2,9.8) | 1.5(1.4,1.7) | 1.5(1.4,1.5) | -(-,-) | -(-,-) | 0.8(0.7,0.9) | 22.2(19.0,26.1) | 581(549,615) | -(-,-) | -(-,-) | 0.17(0.15,0.19) |
| CESM104 | 3.2(3.1,3.2) | 1.6(1.5,1.6) | 1.2(1.1,1.2) | 1.1(1.1,1.1) | -(-,-) | 4.1(3.7,4.5) | 1.2(1.0,1.4) | 1.0(0.8,1.2) | 0.6(0.6,0.7) | -(-,-) | 4.8(4.0,5.6) | 78.1(56.7,94.9) | 326(260,403) | 4092(3138,4824) | -(-,-) | -0.11(-0.16,-0.049) |
| CNRMCM61 | 2.7(2.6,2.9) | 2.7(2.6,2.9) | 3.5(3.5,3.5) | -(-,-) | -(-,-) | 3.5(3.0,4.0) | 1.4(1.2,1.6) | 1.6(1.5,1.7) | -(-,-) | -(-,-) | 3.4(2.6,4.5) | 43.6(33.8,57.9) | 415(384,455) | -(-,-) | -(-,-) | 2.5(2.5,2.5) |
| ECEARTH | 1.0(0.7,1.4) | 4.0(3.7,4.3) | 2.5(2.4,2.5) | -(-,-) | -(-,-) | 3.3(0.7,4.8) | 1.0(0.4,3.3) | 2.4(2.1,2.5) | -(-,-) | -(-,-) | 8.5(3.5,9.8) | 38.0(13.5,89.3) | 269(254,294) | -(-,-) | -(-,-) | -0.6(-0.64,-0.58) |
| ECHAM5MPIOM | 1.7(1.4,1.9) | 4.1(3.8,4.3) | 5.1(5.0,5.1) | -(-,-) | -(-,-) | 3.2(2.1,5.6) | 2.0(1.6,2.4) | 2.5(2.3,2.7) | -(-,-) | -(-,-) | 2.4(1.1,5.4) | 37.4(28.6,50.8) | 744(567,931) | -(-,-) | -(-,-) | 1.9(1.6,2.2) |
| FAMOUS | 3.0(2.9,3.2) | 5.6(5.4,5.7) | 3.6(3.5,3.7) | 1.8(1.8,1.8) | -(-,-) | 3.9(3.4,4.4) | 1.9(1.5,2.4) | 0.4(0.3,0.5) | 0.1(0.0,0.1) | -(-,-) | 5.3(4.0,6.9) | 36.5(28.5,46.4) | 346(239,474) | 3079(1320,4661) | -(-,-) | 0.24(0.19,0.26) |
| GFDLCM3 | 0.4(0.2,0.7) | 2.3(2.1,2.6) | 3.9(3.2,4.4) | 3.4(2.9,4.1) | -(-,-) | 5.6(1.9,9.1) | 1.6(1.2,5) | 2.0(1.7,2.3) | 0.9(0.5,1.3) | -(-,-) | 4.2(1.3,8.0) | 66.4(36.2,91.5) | 852(609,971) | 2865(1723,4397) | -(-,-) | -0.2(-0.28,-0.12) |
| GFDLESM2M | 2.1(2.0,2.2) | 0.00(0.0,0.1) | 3.9(3.7,4.1) | 0.7(0.6,0.9) | -(-,-) | 4.6(1.7,7.8) | 0.6(0.1,1.5) | 2.3(2.0,2.5) | 0.3(0.1,0.5) | -(-,-) | 4.0(1.4,7.6) | 56.0(17.5,91.2) | 392(262,465) | 3230(1562,4572) | -(-,-) | -0.22(-0.28,-0.18) |
| GISSE2R | 2.4(2.4,2.5) | 0.9(0.9,1.0) | 1.4(1.4,1.4) | -(-,-) | -(-,-) | 5.4(4.9,5.9) | 1.4(1.3,1.5) | 1.5(1.4,1.5) | -(-,-) | -(-,-) | 2.1(1.9,2.4) | 92.0(85.0,97.7) | 732(709,754) | -(-,-) | -(-,-) | 0.11(0.1,0.11) |
| HadCM3L | 2.3(2.2,2.5) | 1.8(1.6,1.9) | 1.9(1.9,1.9) | -(-,-) | -(-,-) | 3.9(3.2,4.4) | 0.5(0.2,1.1) | 1.4(1.2,1.5) | -(-,-) | -(-,-) | 5.0(3.7,6.1) | 45.5(13.2,91.5) | 327(277,454) | -(-,-) | -(-,-) | 0.8(0.7,0.85) |
| HadGEM2 | 4.1(3.8,4.5) | 1.2(1.1,1.7) | 1.9(1.6,2.0) | 1.6(0.3,4.3) | 3.7(0.5,7.6) | 4.1(2.8,7.2) | 2.4(1.9,2.7) | 1.3(1.2,1.4) | 0.7(0.1,1.6) | 1.7(0.3,3.2) | 1.3(0.7,2.4) | 11.2(10.2,13.6) | 242(207,289) | 3574(1539,4779) | 7540(5516,9531) | -0.95(-2.1,0.003) |
| IPSLCM5A | 2.4(2.3,2.6) | 2.6(2.5,2.7) | 2.4(2.4,2.5) | -(-,-) | -(-,-) | 2.3(1.9,2.7) | 1.5(1.2,1.9) | 1.3(1.1,1.4) | -(-,-) | -(-,-) | 5.9(3.9,8.3) | 38.0(27.2,59.5) | 343(292,453) | -(-,-) | -(-,-) | 1.1(1.1,1.2) |
| MIROC32 | 4.6(2.1,5.4) | 0.9(0.0,3.4) | 3.1(3.0,3.1) | 0.1(0.0,0.2) | 0.1(0.0,0.02) | 3.9(1.0,8.3) | 1.7(0.4,3.1) | 2.0(1.9,2.1) | 0.2(0.0,0.5) | 0.3(0.1,0.4) | 4.4(0.6,8.6) | 15.8(11.3,26.3) | 586(552,617) | 3125(1493,4693) | 7820(5638,9575) | 1.0(0.96,1.1) |
| MPIESM11 | 2.4(2.3,2.2) | 1.5(1.0,1.8) | 1.7(1.6,1.7) | 1.3(1.3,1.4) | -(-,-) | 4.3(3.4,4.8) | 1.0(0.4,2.5) | 1.4(1.2,1.5) | 1.0(0.8,1.1) | -(-,-) | 5.4(2.5,7.1) | 31.8(12.8,93.2) | 247(203,434) | 1516(1276,2948) | -(-,-) | 0.41(0.31,0.44) |
| MPIESM12 | 2.7(2.5,2.9) | 1.5(1.3,1.7) | 2.2(2.2,2.3) | -(-,-) | -(-,-) | 4.3(3.5,5.2) | 1.8(1.1,2.7) | 1.9(1.8,2.0) | -(-,-) | -(-,-) | 3.1(1.6,4.9) | 21.1(13.7,42.8) | 371(325,449) | -(-,-) | -(-,-) | 0.82(0.72,0.88) |

**Table A1.** Fitted parameters and uncertainties for the LongRunMIP experiments

| Model | Years | $EffCS$ | $\Delta T_{best-est}$ | $\Delta T_{extrap}$ | $R^{4x}_{extrap}$ | $R^{CTRL}_0$ |
|---|---|---|---|---|---|---|
| ACCESS1-0 | 150 | 3.87 | - | 5.6 (3.92,6.82) | 0.33 (-1.21,1.04) | 0.31 |
| ACCESS1-3 | 151 | 3.54 | - | 4.52 (3.14,6.22) | 0.5 (-0.55,1.03) | 0.13 |
| CCSM4 | 104 | 3.98 | - | 2.98 (2.54,3.69) | 1.26 (-0.28,1.98) | -0.01 |
| CNRM-CM5 | 150 | 3.26 | - | 4.18 (3.22,4.79) | 1.16 (0.12,2.15) | 1.84 |
| CSIRO-Mk3-6-0 | 150 | 4.15 | - | 6.12 (4.87,6.71) | 0.22 (-1.13,1.09) | 0.33 |
| CanESM2 | 5 | - | - | 4.48 (3.42,5.6) | -1.23 (-2.39,0.98) | 0.11 |
| FGOALS-s2 | 150 | 4.23 | - | 5.74 (4.16,7.0) | 0.36 (-1.18,1.44) | 0.47 |
| GFDL-CM3 | 150 | 3.94 | - | 3.89 (3.23,5.69) | -0.24 (-2.08,0.75) | 0.18 |
| GFDL-ESM2G | 300 | 2.57 | - | 2.71 (2.52,3.33) | -0.22 (-0.79,0.32) | -0.01 |
| GFDL-ESM2M | 300 | 2.68 | - | 3.87 (3.25,4.41) | -0.18 (-0.91,0.38) | 0.02 |
| GISS-E2-H | 151 | 2.43 | - | 4.19 (3.12,5.07) | -1.15 (-2.15,0.17) | 0.54 |
| GISS-E2-R | 151 | 2.36 | - | 2.73 (2.15,4.44) | -0.5 (-1.99,0.63) | 0.13 |
| HadGEM2-ES | 5 | - | - | 5.58 (3.63,7.22) | -1.34 (-2.41,0.69) | 0.20 |
| IPSL-CM5A-LR | 5 | - | - | 3.23 (3.18,3.99) | -1.14 (-2.37,1.04) | 0.17 |
| IPSL-CM5A-MR | 140 | 4.10 | - | 5.07 (3.56,6.33) | 0.72 (-0.3,1.18) | 0.22 |
| IPSL-CM5B-LR | 160 | 2.63 | - | 3.43 (2.42,4.69) | -0.28 (-1.77,0.5) | 0.14 |
| MIROC-ESM | 150 | 4.65 | - | 4.61 (3.72,6.95) | -1.75 (-2.42,-0.61) | -1.41 |
| MIROC5 | 6 | - | - | 2.79 (2.38,3.88) | -1.45 (-2.42,0.33) | -0.37 |
| MPI-ESM-LR | 150 | 3.63 | - | 6.09 (4.34,7.19) | -0.23 (-1.84,0.91) | 0.27 |
| MPI-ESM-P | 150 | 3.45 | - | 5.86 (3.79,7.03) | -0.46 (-1.99,0.78) | 0.28 |
| NorESM1-M | 150 | 2.80 | - | 4.6 (3.12,5.58) | 1.03 (-0.12,1.83) | 1.12 |
| bcc-csm1-1-m | 150 | 2.82 | - | 3.29 (2.76,5.23) | -1.1 (-2.27,-0.14) | -0.35 |
| inmcm4 | 150 | 2.04 | - | 1.92 (1.65,2.16) | 0.36 (-0.3,0.96) | 0.36 |

**Table A2.** Fitted parameters and uncertainties for the CMIP5 experiments

| Model | Years | $EffCS$ | $\Delta T_{best-est}$ | $\Delta T_{extrap}$ | $R^{4x}_{extrap}$ | $R^{CTRL}_0$ |
|---|---|---|---|---|---|---|
| ACCESS-CM2 | 150 | 4.70 | - | 3.61 (3.4,4.28) | -0.28 (-1.5,0.98) | 0.25 |
| AWI-CM-1-1-MR | 151 | 3.13 | - | 4.47 (3.18,5.82) | -0.25 (-1.83,0.6) | 0.20 |
| BCC-CSM2-MR | 151 | 2.98 | - | 5.48 (3.62,6.61) | -1.43 (-2.38,-0.02) | -0.21 |
| CAMS-CSM1-0 | 150 | 2.30 | - | 3.52 (2.51,4.38) | -0.3 (-1.83,0.77) | 0.59 |
| CESM2 | 400 | 6.08 | - | 5.45 (5.29,5.72) | 0.36 (-0.21,0.88) | 0.41 |
| CESM2-WACCM | 150 | 4.71 | - | 6.59 (5.47,7.15) | -0.24 (-1.86,0.86) | 0.10 |
| CIESM | 150 | 5.65 | - | 7.25 (6.08,7.86) | 1.93 (0.7,2.24) | 1.19 |
| CMCC-CM2-SR5 | 165 | 3.59 | - | 4.84 (3.6,6.13) | 0.78 (-0.89,1.66) | 1.15 |
| CNRM-CM6-1 | 150 | 4.90 | - | 5.36 (3.97,6.54) | 0.83 (-1.05,1.69) | 0.78 |
| CNRM-CM6-1-HR | 150 | 4.38 | - | 6.25 (4.56,7.21) | -0.07 (-1.5,1.34) | 0.88 |
| CNRM-ESM2-1 | 150 | 4.72 | - | 3.5 (3.24,4.49) | 0.35 (-1.07,1.46) | 0.79 |
| CanESM5 | 151 | 5.59 | - | 6.49 (4.77,7.96) | 0.1 (-1.47,1.01) | 0.11 |
| EC-Earth3 | 160 | 4.29 | - | 4.42 (3.74,5.75) | 0.27 (-0.82,0.62) | 0.09 |
| FGOALS-f3-L | 160 | 3.02 | - | 4.35 (3.06,5.68) | -0.31 (-1.96,0.67) | 0.20 |
| FGOALS-g3 | 152 | 2.87 | - | 3.35 (2.54,4.56) | 0.97 (-0.44,1.63) | 0.99 |
| FIO-ESM-2-0 | 150 | 4.24 | - | 6.69 (5.1,7.33) | -0.61 (-2.02,1.27) | 1.13 |
| GFDL-ESM4 | 150 | 2.66 | - | 3.04 (2.32,3.81) | 0.06 (-1.26,0.76) | 0.12 |
| GISS-E2-1-G | 151 | 2.72 | - | 4.02 (3.17,4.53) | -0.26 (-1.45,0.74) | 0.09 |
| HadGEM3-GC31-LL | 150 | 5.56 | - | 6.76 (5.14,7.96) | 0.11 (-1.4,1.02) | 0.15 |
| HadGEM3-GC31-MM | 150 | 5.47 | - | 7.24 (6.01,7.75) | -0.48 (-2.11,0.88) | 0.20 |
| IITM-ESM | 165 | 2.38 | - | 2.89 (2.22,3.36) | 0.5 (-0.86,1.16) | 0.44 |
| INM-CM4-8 | 150 | 1.83 | - | 3.27 (2.29,3.99) | 0.95 (-0.01,2.02) | 2.17 |
| INM-CM5-0 | 150 | 1.91 | - | 3.13 (2.08,3.67) | 0.27 (-0.74,1.04) | 0.88 |
| KACE-1-0-G | 151 | 4.97 | - | 6.37 (5.01,7.17) | 0.99 (-0.51,1.61) | 0.56 |
| MIROC-ES2L | 150 | 2.66 | - | 2.54 (2.15,3.02) | 0.69 (-0.61,1.34) | 0.48 |
| MIROC6 | 250 | 2.62 | - | 2.97 (2.4,3.36) | 0.8 (0.07,1.29) | 0.60 |
| MPI-ESM1-2-HR | 165 | 2.97 | - | 3.62 (2.8,4.6) | -0.49 (-1.53,0.65) | 0.16 |
| MPI-ESM1-2-LR | 165 | 3.04 | - | 4.44 (3.25,5.19) | -0.5 (-1.64,0.72) | 0.26 |
| NESM3 | 150 | 4.69 | - | 5.24 (4.16,6.06) | 0.21 (-0.96,0.82) | -0.06 |
| NorESM2-MM | 150 | 2.43 | - | 3.43 (2.46,4.32) | 0.03 (-1.51,0.75) | 0.05 |
| TaiESM1 | 150 | 4.36 | - | 6.93 (5.48,7.45) | -0.85 (-2.09,0.53) | 0.08 |
| UKESM1-0-LL | 150 | 5.37 | - | 6.13 (4.62,7.55) | 0.33 (-1.31,0.9) | 0.05 |

**Table A3.** Fitted parameters and uncertainties for the CMIP6 experiments

**Code and data availability.** All code to reproduce this study is available at https://doi.org/10.5281/zenodo.6424714. CMIP5 and CMIP6 source data is freely available, and was here accessed on the Google Public Cloud https://console.cloud.google.com/storage/browser/cmip6. Longrunmip data is available on request at http://www.longrunmip.org/

**Author contributions.** BMS performed all calculations, produced plots and wrote main text. MR produced ESM data and co-wrote main text. [5]

**Competing interests.** The authors have no competing interests

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

| Model | $S_1$ | $S_2$ | $S_3$ | $S_4$ | $S_5$ | $R_1$ | $R_2$ | $R_3$ | $R_4$ | $R_5$ | $\tau_1$ | $\tau_2$ | $\tau_3$ | $\tau_4$ | $\tau_5$ | $R_{extrap}^{4x}$ |
|---|---|---|---|---|---|---|---|---|---|---|---|---|---|---|---|---|
| ACCESS1-0 | 2.7(2.4,3.0) | 1.2(1.0,1.4) | 7.3(4.5,9.4) | -(-,-) | -(-,-) | 3.5(2.9,4.2) | 1.2(0.5,1.7) | 2.2(0.5,4.9) | -(-,-) | -(-,-) | 4.0(2.7,5.7) | 43.6(18.1,83.5) | 668(273,940) | -(-,-) | -(-,-) | 0.66(-1.8,1.9) |
| ACCESS1-3 | 3.0(2.7,3.1) | 1.1(0.8,1.8) | 5.0(2.2,8.1) | -(-,-) | -(-,-) | 3.5(2.9,4.1) | 1.5(1.0,2.1) | 1.5(0.3,3.4) | -(-,-) | -(-,-) | 3.1(1.9,4.8) | 22.9(12.7,44.5) | 672(267,942) | -(-,-) | -(-,-) | 1.0(-0.7,2.0) |
| CCSM4 | 2.6(2.4,3.0) | 2.1(1.8,2.3) | 1.3(0.4,2.4) | -(-,-) | -(-,-) | 3.8(2.5,4.3) | 1.0(0.3,2.2) | 1.9(0.4,4.5) | -(-,-) | -(-,-) | 6.1(4.1,7.6) | 29.1(12.3,82.3) | 698(322,945) | -(-,-) | -(-,-) | 2.5(0.14,3.8) |
| CNRM-CM5 | 2.0(1.7,2.2) | 2.6(2.4,2.8) | 3.8(2.3,4.9) | -(-,-) | -(-,-) | 3.5(2.9,4.3) | 2.2(1.8,2.7) | 3.4(1.8,5.2) | -(-,-) | -(-,-) | 1.9(1.2,3.1) | 16.2(12.4,23.7) | 700(334,944) | -(-,-) | -(-,-) | 2.3(0.64,3.9) |
| CSIRO-Mk3-6-0 | 2.0(1.7,2.5) | 1.5(1.2,1.8) | 8.7(6.7,9.8) | -(-,-) | -(-,-) | 5.1(3.5,7.8) | 1.6(1.0,2.1) | 2.5(0.6,4.8) | -(-,-) | -(-,-) | 1.5(0.9,3.2) | 21.6(12.4,44.2) | 702(286,950) | -(-,-) | -(-,-) | 0.44(-1.7,2.0) |
| CanESM2 | 3.5(3.4,3.7) | 1.7(1.6,2.0) | 3.7(1.8,5.6) | -(-,-) | -(-,-) | 5.7(3.1,8.3) | 2.6(0.4,6.5) | 2.7(0.4,6.4) | -(-,-) | -(-,-) | 2.2(0.9,6.6) | 54.6(19.6,92.0) | 544(184,899) | -(-,-) | -(-,-) | -2.5(-4.5,1.0) |
| FGOALS-s2 | 4.5(4.3,4.5) | 0.3(0.1,0.7) | 6.7(4.0,9.0) | -(-,-) | -(-,-) | 3.9(3.1,4.7) | 0.8(0.2,1.5) | 3.2(0.8,5.9) | -(-,-) | -(-,-) | 4.3(2.6,5.9) | 43.1(12.2,89.4) | 697(293,945) | -(-,-) | -(-,-) | 0.72(-1.7,2.7) |
| GFDL-CM3 | 2.9(2.8,3.0) | 2.6(1.9,3.2) | 2.3(0.5,5.4) | -(-,-) | -(-,-) | 4.1(3.4,5.4) | 1.5(0.9,2.0) | 3.1(0.6,6.5) | -(-,-) | -(-,-) | 2.8(1.5,4.0) | 45.0(14.8,82.0) | 651(292,937) | -(-,-) | -(-,-) | -0.48(-3.5,1.3) |
| GFDL-ESM2G | 3.0(3.0,3.1) | 0.1(0.0,0.2) | 2.3(2.0,3.0) | -(-,-) | -(-,-) | 3.5(3.0,4.0) | 0.5(0.1,1.1) | 2.2(1.2,3.2) | -(-,-) | -(-,-) | 5.5(3.9,6.8) | 47.5(13.3,91.9) | 697(314,948) | -(-,-) | -(-,-) | -0.45(-1.4,0.5) |
| GFDL-ESM2M | 2.9(2.9,3.0) | 0.1(0.0,0.2) | 4.7(3.7,5.6) | -(-,-) | -(-,-) | 3.7(3.1,4.3) | 0.9(0.5,1.4) | 2.3(1.4,3.6) | -(-,-) | -(-,-) | 3.9(2.4,5.3) | 41.4(14.8,88.4) | 625(236,940) | -(-,-) | -(-,-) | -0.36(-1.6,0.62) |
| GISS-E2-H | 2.1(2.1,2.2) | 1.6(1.5,1.7) | 4.6(2.8,6.2) | -(-,-) | -(-,-) | 3.5(3.0,4.1) | 2.3(1.9,2.6) | 4.7(2.6,6.5) | -(-,-) | -(-,-) | 1.9(1.4,2.6) | 13.5(1.2,17.2) | 695(325,948) | -(-,-) | -(-,-) | -2.3(-4.0,-0.095) |
| GISS-E2-R | 2.0(1.9,2.2) | 0.9(0.7,1.0) | 2.6(1.6,5.5) | -(-,-) | -(-,-) | 5.0(4.2,6.1) | 1.6(1.3,1.9) | 3.6(1.3,6.1) | -(-,-) | -(-,-) | 1.7(1.1,2.3) | 23.5(13.8,42.9) | 664(256,940) | -(-,-) | -(-,-) | -1.0(-3.4,0.96) |
| HadGEM2-ES | 4.2(1.7,4.4) | 1.8(0.9,3.2) | 5.9(1.4,9.0) | -(-,-) | -(-,-) | 3.6(2.0,6.0) | 2.9(0.5,6.6) | 2.8(0.5,6.1) | -(-,-) | -(-,-) | 3.3(0.8,8.2) | 58.4(20.5,92.9) | 574(94,910) | -(-,-) | -(-,-) | -2.7(-4.6,0.57) |
| IPSL-CM5A-LR | 2.2(1.9,2.5) | 2.4(2.2,2.6) | 2.0(1.9,2.2) | -(-,-) | -(-,-) | 2.6(1.6,4.5) | 2.8(0.5,6.9) | 2.7(0.5,6.6) | -(-,-) | -(-,-) | 2.6(0.7,7.8) | 48.2(18.8,88.4) | 561(188,912) | -(-,-) | -(-,-) | -2.3(-4.5,1.2) |
| IPSL-CM5A-MR | 2.2(1.8,2.6) | 2.7(2.4,2.9) | 5.2(2.8,7.4) | -(-,-) | -(-,-) | 2.3(1.6,2.9) | 2.3(1.7,2.8) | 1.3(0.2,3.0) | -(-,-) | -(-,-) | 3.5(1.8,6.0) | 21.6(15.4,31.0) | 661(268,933) | -(-,-) | -(-,-) | 1.4(-0.13,2.3) |
| IPSL-CM5B-LR | 2.1(1.9,2.3) | 1.4(1.2,1.7) | 3.3(1.4,5.5) | -(-,-) | -(-,-) | 3.4(2.5,4.9) | 1.4(0.7,1.8) | 2.4(0.6,4.9) | -(-,-) | -(-,-) | 2.8(1.4,4.9) | 36.3(16.8,78.8) | 684(309,945) | -(-,-) | -(-,-) | -0.57(-2.8,0.85) |
| MIROC-ESM | 2.4(2.1,2.9) | 2.8(2.5,3.1) | 3.9(2.5,7.9) | -(-,-) | -(-,-) | 6.5(5.1,8.7) | 2.9(2.4,3.3) | 4.2(2.3,5.4) | -(-,-) | -(-,-) | 1.2(0.9,1.8) | 11.2(10.2,14.4) | 578(296,799) | -(-,-) | -(-,-) | -3.5(-4.7,-1.7) |
| MIROC5 | 3.5(3.2,3.6) | 0.9(0.6,1.1) | 1.3(0.3,3.3) | -(-,-) | -(-,-) | 5.3(2.6,7.7) | 3.4(0.6,7.6) | 2.4(0.4,5.2) | -(-,-) | -(-,-) | 6.1(2.4,9.1) | 45.4(14.5,87.5) | 540(174,909) | -(-,-) | -(-,-) | -2.9(-4.6,-0.13) |
| MPI-ESM-LR | 2.9(2.6,3.4) | 1.7(1.3,2.0) | 7.6(4.7,9.6) | -(-,-) | -(-,-) | 4.6(3.6,5.4) | 1.9(1.0,3.0) | 3.4(0.8,6.3) | -(-,-) | -(-,-) | 3.6(1.7,5.8) | 23.3(1.8,70.5) | 700(293,951) | -(-,-) | -(-,-) | -0.45(-3.2,1.6) |
| MPI-ESM-P | 2.9(2.6,3.2) | 1.7(1.4,1.9) | 7.2(3.9,9.2) | -(-,-) | -(-,-) | 5.0(4.0,6.3) | 2.2(1.4,3.0) | 3.8(1.4,6.6) | -(-,-) | -(-,-) | 2.4(1.4,4.3) | 17.7(11.6,49.1) | 676(294,934) | -(-,-) | -(-,-) | -0.92(-3.4,1.2) |
| NorESM1-M | 1.9(1.7,2.2) | 1.2(1.0,1.4) | 6.0(3.7,7.7) | -(-,-) | -(-,-) | 3.7(3.0,4.5) | 1.5(0.9,2.1) | 2.4(0.7,4.4) | -(-,-) | -(-,-) | 2.7(1.5,4.3) | 19.9(1.7,47.5) | 696(293,950) | -(-,-) | -(-,-) | 2.1(0.19,3.5) |
| bcc-csm1-1-m | 2.2(2.0,2.5) | 1.8(1.6,2.1) | 2.5(1.7,5.7) | -(-,-) | -(-,-) | 4.0(3.0,4.6) | 1.5(0.6,2.6) | 3.0(0.9,5.2) | -(-,-) | -(-,-) | 5.3(2.6,7.2) | 31.8(13.0,81.1) | 636(270,927) | -(-,-) | -(-,-) | -2.2(-4.2,-0.5) |
| inmcm4 | 2.2(2.0,2.8) | 0.6(0.1,0.9) | 1.0(0.6,1.4) | -(-,-) | -(-,-) | 3.4(3.1,3.7) | 0.2(0.0,0.6) | 1.9(0.8,3.0) | -(-,-) | -(-,-) | 4.5(3.7,5.1) | 49.6(13.1,90.2) | 724(377,949) | -(-,-) | -(-,-) | 0.72(-0.34,1.7) |

**Table A4.** Fitted parameters and uncertainties for the CMIP5 experiments

| Model | $S_1$ | $S_2$ | $S_3$ | $S_4$ | $S_5$ | $R_1$ | $R_2$ | $R_3$ | $R_4$ | $R_5$ | $\tau_1$ | $\tau_2$ | $\tau_3$ | $\tau_4$ | $\tau_5$ | $R^{4x}_{estrap}$ |
|---|---|---|---|---|---|---|---|---|---|---|---|---|---|---|---|---|
| ACCESS-CM2 | 3.0(2.9,3.1) | 3.6(3.4,3.7) | 0.7(0.1,1.8) | (-) | (-) | 5.1(3.7,7.2) | 2.6(2.0,2.9) | 3.9(1.6,6.0) | (-) | (-) | 1.2(0.8,2.2) | 13.5(10.9,20.4) | 652(228,943) | (-) | (-) | -0.57(-2.6,1.6) |
| AWI-CM-1-1-MR | 3.1(2.6,3.5) | 1.5(1.2,1.8) | 4.3(2.1,6.7) | (-) | (-) | 4.3(3.3,5.2) | 1.8(0.9,2.7) | 2.5(0.5,5.3) | (-) | (-) | 3.5(1.7,6.1) | 27.9(12.9,71.1) | 686(280,943) | (-) | (-) | -0.5(-3.0,0.99) |
| BCC-CSM2-MR | 1.9(1.7,2.4) | 1.7(1.3,1.9) | 7.4(4.4,9.4) | (-) | (-) | 3.8(2.9,6.2) | 2.1(1.2,2.5) | 4.6(2.0,6.3) | (-) | (-) | 1.4(0.8,4.8) | 14.0(10.8,52.1) | 654(308,898) | (-) | (-) | -2.9(-4.5,6.39) |
| CAMS-CSM1-0 | 2.2(2.1,2.5) | 1.3(1.1,1.4) | 3.5(1.8,5.0) | (-) | (-) | 5.2(3.8,7.5) | 2.2(1.2,3.0) | 3.2(1.1,5.9) | (-) | (-) | 2.0(1.0,4.4) | 17.2(11.3,45.8) | 704(322,947) | (-) | (-) | -0.61(-3.0,1.3) |
| CESM2 | 4.3(4.2,4.3) | 1.4(0.6,2.0) | 5.3(5.0,5.8) | (-) | (-) | 5.5(4.9,6.0) | 1.1(0.7,1.4) | 2.6(2.0,3.5) | (-) | (-) | 3.7(3.0,4.3) | 60.6(28.4,90.0) | 620(255,944) | (-) | (-) | 0.73(-0.2,1.7) |
| CESM2-WACCM | 3.6(3.0,4.1) | 0.8(0.3,1.2) | 8.8(7.1,9.8) | (-) | (-) | 5.2(4.6,5.9) | 0.9(0.4,1.5) | 3.5(1.0,6.4) | (-) | (-) | 3.3(2.5,4.0) | 51.0(13.7,91.0) | 668(282,933) | (-) | (-) | -0.49(-3.1,1.5) |
| CIESM | 3.6(3.3,3.8) | 2.1(1.9,2.3) | 8.9(7.2,9.8) | (-) | (-) | 3.9(3.5,4.3) | 2.3(1.7,2.6) | 0.9(0.1,2.9) | (-) | (-) | 4.9(3.9,6.0) | 54.6(41.0,73.3) | 658(205,942) | (-) | (-) | 3.9(2.4,4.4) |
| CMCC-CM2-SR5 | 3.3(3.0,3.5) | 1.8(1.6,2.0) | 4.6(2.5,6.8) | (-) | (-) | 3.5(2.2,4.2) | 1.2(0.4,2.3) | 2.6(0.5,5.5) | (-) | (-) | 6.7(3.8,9.1) | 44.7(13.6,87.5) | 696(296,942) | (-) | (-) | 1.6(-1,3.2) |
| CNRM-CM6-1 | 3.0(2.8,3.2) | 2.8(2.6,2.9) | 4.9(2.5,6.9) | (-) | (-) | 3.4(2.8,4.0) | 1.5(1.0,1.9) | 2.7(0.6,5.9) | (-) | (-) | 3.2(2.0,4.7) | 34.2(15.5,73.3) | 679(274,939) | (-) | (-) | 1.7(-1,3.2) |
| CNRM-CM6-1-HR | 2.1(1.9,2.3) | 3.2(3.0,3.4) | 7.1(4.4,8.8) | (-) | (-) | 3.0(2.3,4.1) | 2.4(2.0,2.7) | 4.7(2.3,7.1) | (-) | (-) | 1.9(1.1,3.0) | 19.9(15.5,28.5) | 686(284,945) | (-) | (-) | -0.14(-2.5,2.2) |
| CNRM-ESM2-1 | 4.1(3.9,4.3) | 2.0(1.6,2.3) | 0.9(0.2,2.6) | (-) | (-) | 1.8(0.4,3.4) | 1.2(0.3,2.2) | 3.6(1.4,6.0) | (-) | (-) | 7.8(0.2,9.6) | 20.8(11.3,76.4) | 669(310,941) | (-) | (-) | 0.7(-1.5,2.6) |
| CanESM5 | 2.7(2.3,3.0) | 3.5(3.2,3.7) | 6.8(4.1,9.4) | (-) | (-) | 2.5(2.0,3.0) | 2.0(1.6,2.4) | 2.7(0.6,5.5) | (-) | (-) | 4.9(3.2,7.0) | 31.4(19.3,54.2) | 656(281,942) | (-) | (-) | 0.19(-2.3,1.8) |
| EC-Earth3 | 3.4(3.3,3.6) | 3.2(3.0,3.6) | 2.2(0.7,4.5) | (-) | (-) | 4.6(3.8,5.6) | 2.1(1.7,2.5) | 1.1(0.2,3.7) | (-) | (-) | 2.7(2.0,3.7) | 44.3(30.9,60.1) | 660(228,945) | (-) | (-) | 0.54(-1,1.2) |
| FGOALS-f3-L | 2.8(2.4,3.3) | 1.3(1.0,1.7) | 4.5(2.3,6.8) | (-) | (-) | 5.1(4.2,5.9) | 1.4(0.5,2.3) | 3.0(0.7,6.0) | (-) | (-) | 4.3(2.6,5.7) | 35.7(12.6,88.5) | 684(290,938) | (-) | (-) | -0.62(-3.3,1.1) |
| FGOALS-g3 | 3.2(3.1,3.3) | 0.8(0.6,1.3) | 2.6(0.7,4.7) | (-) | (-) | 4.1(3.7,4.4) | 1.1(0.8,1.5) | 3.9(3.2,4.6) | (-) | (-) | 3.9(3.1,5.8) | 38.3(15.8,71.0) | 671(269,946) | (-) | (-) | 1.9(-0.37,3.1) |
| FIO-ESM-2-0 | 2.1(1.9,4.5) | 2.6(0.3,2.9) | 8.6(6.1,9.8) | (-) | (-) | 4.6(3.8,5.4) | 0.8(0.2,1.6) | 6.6(3.3,9.3) | (-) | (-) | 2.8(2.2,3.5) | 69.7(25.1,94.7) | 641(312,905) | (-) | (-) | -1.2(-3.7,1.19) |
| GFDL-ESM4 | 2.6(2.3,3.0) | 1.1(0.8,1.4) | 2.3(1.1,3.7) | (-) | (-) | 4.0(3.1,5.7) | 2.0(0.4,4.3) | 2.8(2.2,3.5) | (-) | (-) | 3.4(1.2,5.6) | 27.2(11.7,77.9) | 711(287,944) | (-) | (-) | 0.13(-2.0,1.4) |
| GISS-E2-1-G | 3.4(3.4,3.5) | 0.0(0.0,0.2) | 4.6(3.2,5.4) | (-) | (-) | 3.8(3.2,4.4) | 0.5(0.1,1.1) | 3.2(1.3,5.3) | (-) | (-) | 4.2(2.9,5.2) | 50.6(12.2,90.7) | 716(339,958) | (-) | (-) | -0.53(-2.5,1.1) |
| HadGEM3-GC31-LL | 3.1(2.9,3.3) | 3.5(3.4,3.7) | 6.8(4.3,8.9) | (-) | (-) | 3.4(3.0,3.9) | 2.1(1.8,2.4) | 2.7(0.8,5.3) | (-) | (-) | 2.9(2.1,3.9) | 28.5(19.4,42.2) | 669(264,948) | (-) | (-) | 0.21(-2.1,1.8) |
| HadGEM3-GC31-MM | 2.2(2.0,2.3) | 3.4(3.2,3.5) | 8.9(7.1,9.8) | (-) | (-) | 3.8(3.1,5.0) | 2.0(1.7,2.4) | 4.2(1.4,7.2) | (-) | (-) | 1.9(1.2,2.7) | 29.9(20.6,46.4) | 731(341,945) | (-) | (-) | -0.97(-3.7,1.4) |
| IITM-ESM | 2.5(2.3,2.8) | 0.9(0.7,1.1) | 2.3(1.2,3.1) | (-) | (-) | 4.3(3.2,5.6) | 1.3(0.5,2.4) | 2.0(0.5,4.3) | (-) | (-) | 3.7(1.4,5.7) | 29.7(12.0,79.1) | 701(296,952) | (-) | (-) | 1.0(-1,2.2) |
| INM-CM4-8 | 1.7(1.6,1.9) | 1.2(1.0,1.3) | 3.7(2.0,4.9) | (-) | (-) | 3.5(2.8,4.5) | 1.9(1.4,2.3) | 3.4(1.6,5.0) | (-) | (-) | 1.8(1.1,3.0) | 13.0(10.6,20.9) | 672(292,943) | (-) | (-) | 1.9(0.29,3.6) |
| INM-CM5-0 | 1.8(1.7,2.6) | 1.1(0.8,1.2) | 3.4(1.5,4.3) | (-) | (-) | 3.5(2.8,4.0) | 1.3(0.7,2.1) | 2.3(0.8,4.0) | (-) | (-) | 3.4(1.8,5.2) | 19.1(11.3,48.1) | 695(311,950) | (-) | (-) | 0.54(-1.0,1.9) |
| KACE-1-0-G | 3.3(3.0,3.6) | 1.2(1.0,1.5) | 8.3(6.1,9.7) | (-) | (-) | 3.3(2.9,3.6) | 1.4(1.0,1.8) | 1.9(0.4,4.6) | (-) | (-) | 4.4(3.5,5.4) | 47.0(22.3,75.3) | 648(239,935) | (-) | (-) | 2.0(-0.3,3.1) |
| MIROC-ES2L | 2.2(1.8,2.8) | 1.6(1.1,2.0) | 1.2(0.5,2.1) | (-) | (-) | 4.2(3.1,6.1) | 1.5(0.5,2.6) | 1.8(0.4,4.1) | (-) | (-) | 3.1(1.2,5.7) | 21.4(11.9,73.2) | 703(296,955) | (-) | (-) | 1.4(-0.6,2.6) |
| MIROC6 | 2.5(2.1,2.8) | 1.1(0.8,1.4) | 2.3(1.4,3.0) | (-) | (-) | 4.1(3.1,5.2) | 1.7(0.6,2.9) | 1.7(0.6,2.9) | (-) | (-) | 3.5(1.9,5.5) | 32.3(12.4,79.1) | 677(277,943) | (-) | (-) | 1.6(0.48,2.5) |
| MPI-ESM1-2-HR | 3.1(2.9,3.1) | 1.2(1.1,1.5) | 3.0(1.5,4.7) | (-) | (-) | 3.9(3.0,4.8) | 2.3(1.6,2.8) | 3.4(1.4,5.3) | (-) | (-) | 1.4(1.1,2.0) | 14.4(11.0,29.2) | 669(265,941) | (-) | (-) | -0.97(-2.7,1.0) |
| MPI-ESM1-2-LR | 2.7(2.5,3.1) | 1.4(1.1,1.7) | 4.7(2.8,6.1) | (-) | (-) | 5.4(4.5,6.5) | 1.9(1.2,2.7) | 3.9(1.8,5.9) | (-) | (-) | 2.1(1.3,3.4) | 13.5(10.6,31.8) | 676(283,943) | (-) | (-) | -1.0(-2.9,1.1) |
| NESM3 | 3.6(3.5,3.7) | 2.6(2.5,2.7) | 4.2(2.4,5.6) | (-) | (-) | 7.4(5.5,9.4) | 2.0(1.8,2.3) | 1.8(0.4,3.8) | (-) | (-) | 0.9(0.7,1.1) | 28.7(22.6,36.3) | 668(279,935) | (-) | (-) | 0.41(-1.4,1.5) |
| NorESM2-MM | 3.2(3.1,3.3) | 0.1(0.0,0.5) | 3.5(1.8,5.0) | (-) | (-) | 5.6(3.8,7.8) | 0.6(0.1,1.3) | 2.0(0.0,4.6) | (-) | (-) | 2.5(1.6,4.1) | 54.1(17.2,91.0) | 674(279,938) | (-) | (-) | 0.055(-2.3,1.13) |
| TaiESM1 | 2.2(2.0,4.5) | 2.7(1.4,3.0) | 8.9(6.7,9.8) | (-) | (-) | 4.7(4.0,5.8) | 2.1(1.5,2.6) | 4.9(2.5,7.0) | (-) | (-) | 1.9(1.3,2.8) | 13.4(10.7,20.3) | 676(232,940) | (-) | (-) | -1.7(-3.8,0.62) |
| UKESM1-0-LL | 3.3(3.1,3.5) | 3.3(3.1,3.4) | 5.6(3.2,8.1) | (-) | (-) | 2.9(2.5,3.3) | 1.8(1.3,2.1) | 1.9(0.4,4.8) | (-) | (-) | 5.3(3.9,6.9) | 44.9(24.0,70.7) | 660(252,940) | (-) | (-) | 0.65(-1.9,1.7) |

**Table A5.** Fitted parameters and uncertainties for the CMIP6 experiments