# Peer review of "Potential for bias in effective climate sensitivity from state-dependent energetic imbalance"

_EGUsphere, 2022_

## Author Comment (AC1)

**Response to reviewer 1**

*General Comments:*

*Sanderson et al. use a simple climate model composed of several exponential decay terms to model the output of pre-industrial control and abrupt-4×CO$_2$ simulations from CMIP5, CMIP6, and LongRunMIP. The authors use this simple climate model to estimate potential biases in effective climate sensitivity (EffCS) estimates. This approach is novel and provides an interesting framework to analyze EffCS; however, there are several points that the authors should address in order for me to recommend this manuscript for publication.*

Many thanks for the careful review and recommendations.

*Specific Comments:*

*1) The new framework the authors developed is interesting; however, I am having a hard time deciphering why this paper is important. I am not sure what the main point of the paper is. Is the main point to answer the question the authors stated at the end of the introduction: "How plausible are the higher sensitivity [CMIP6] models"? Is the main point to say that ECS is actually higher than suggested by EffCS given by CMIP6 or IPCC AR6? The authors state "Our results highlight the potential for error in estimates of effective climate sensitivity through the assumptions on the asymptotic radiative balance of climate models (page 9 line 9)". The authors need to go a step further and provide an indication of what their suggestion for the value of EffCS would be based on their new framework. The authors should discuss their results in the context of recent literature that examines estimates of EffCS. Recent studies have provided estimates of EffCS, such as Zelinka et al. (2020), Tokarska et al. (2020), McBride et al. (2021), Sherwood et al. (2020), and the new comprehensive evaluation conducted by IPCC AR6.*

*Do the authors have a new range of EffCS using their approach compared to these other analyses? Could the authors suggest a way to constrain the estimate of EffCS based on the model's radiative imbalance between the PICTRL and ABRUPT4X simulation? The authors should add comparisons to recent literature in their results section. In the conclusions section, the authors should expand upon the importance of their results to indicate a revision or addition to current estimates of EffCS, or suggestions on how to revise the current estimate of EffCS using their approach.*

Thanks for this point.  We agree that the introduction perhaps could better clarify the objectives of the paper.  The study does not seek to provide a new estimate of real-world climate sensitivity.  This is a somewhat orthogonal issue to the central finding in the study that estimates of model climate sensitivity based on the Gregory regression approach are likely to be biased by different stable energetic states under pre-industrial carbon dioxide concentrations and quadrupled $CO_2$ concentrations.  On revision, we will clarify in the introduction that this study is a comment on the standard approach for calculating effective climate sensitivity, and not on the likelihood of any given value.

We do, however, now highlight that some aspects of the 'hot model' problem may be resolved by the issues raised in this study - in that the model current estimates of Effective Climate Sensitivity are in many cases not reliable.  Quantifying this bias in CMIP6 is not possible with existing simulations, but for LongrunMIP - this bias results in a consistent over-estimate of EffCS in 5 of 15 models (with no examples of the opposite).

We expect that the corrections implied by this study will have some impacts on our assessment of likely ranges of ECS - especially for emergent constraint studies which have identified relationships between published ECS values and observable quantities.  On revision, we will consider to what degree existing literature (including the AR6 assessment) could be impacted by the findings of this study.  A revised assessment of ECS is, however, out of scope for this study.

*2) How did the authors determine the minimum and maximum values of τ for the short timescale, intermediate time scale, and long-time scale given in Table 1? Are these values supported by literature?*

Our priors on timescales are informed by previous studies considering a 3 timescale decomposition of Earth system transient response to abrupt 4xCO2 forcing (e.g.  Proistosescu, C., & Huybers, P. J. (2017), Rugenstein and Armour 2021, Caldeira and Myhrvold ([2013](https://))), noting that the FaIR model used in the IPCC AR6 assessment allows for only 2 timescales (Smith 2018).  There is no universal approach in these former studies for priors on the different timescales. Given this, we here choose relatively uninformative priors compared to previous studies, broadly in place to order to structure the response modes into subdecadal, decadal, centennial, millennial and multi-millennial modes without being overly prescriptive.  On revision, we will clarify our logic on this point.  We also now consider a sensitivity study of how allowance of different timescales in the model impacts our results.

*3) The authors should explain how assessing the radiative imbalance in the control simulation , R0CTRL, impacts the parameters in Eq. 1a or Eq. 1b. As currently written, it is unclear how this assessment is incorporated into Eq. 1a and 1b.*

The control simulation radiative imbalance does not directly impact equations 1a,b.  It is assessed independently from the control simulation and compared with the asymptotic energetic imbalance in

the abrupt4x simulation.  We have now formulated R4x$_{extrap}$ (the equilibrium TOA balance state following the abrupt4x perturbation) to be a parameter in the fitted equation as suggested.  We agree that this is a more accurate way of presenting our approach (it makes no difference to the actual derived fits, which were effectively always derived using  R4x$_{extrap}$).

$$\quad T(t) = \sum_{n=1}^{3} S_n(1 - e^{-(t/\tau_n)}) + T_0$$

$$R(t) = \sum_{n=1}^{3} R_n(-e^{-(t/\tau_n)}) + R_{extrap}^{4x}$$

*4) Equilibrium climate sensitivity and effective climate sensitivity are the response of the climate system to a doubling of CO2 relative to preindustrial. The authors use the ABRUPT4X scenario, which is for a quadrupling of CO2. In other methods, such as Gregory et al. (2004), the temperature response to the quadrupling of CO2 needs to be divided by 2 to achieve an estimate of the temperature response to the doubling of CO2. The authors do not discuss how their method accounts for the fact they are using an ABRUPT4X scenario to assess the temperature response to a doubling of CO2. The authors should elaborate in the methods section how they account for this discrepancy.*

 We have added the following to explain our policy on reporting equilibrium climate sensitivity:

*We report climate sensitivities for a doubling of carbon dioxide from pre-industrial levels. As such, we follow standard practice in dividing ABRUPT4X sensitivities (EffCS, ΔTextrap and ΔTbest−est) by 2 (Meehl et al., 2020), though we note that in some models this approximation introduces minor errors (Jonko et al. (2012); Rugenstein et al. (2019)), this is not the focus of the present study.*

*5) There is no mention of IPCC AR6 in this paper. How does this analysis compare to the best estimate (3°C) and range of (2 - 5°C) of ECS given by AR6? Does the new framework in this paper support a lower or higher value of EffCS than provided by IPCC?*

 Our paper does not contain an estimate of real-world climate sensitivity, rather it is a comment that existing estimates of model climate sensitivity may be unreliable.  This does however have

implications for the use of Effective Climate Sensitivity to assess the reliability of a given model. We now discuss this both at the end of the introduction, and in the conclusions, with explicit reference to AR6 (which was not final when this paper was first drafted).

We have also added a longer discussion on the implications of this study for future assessment:

*This directly impacts our ability to accurately measure $EffCS$ from short simulations, and draws into question whether $EffCS$ should be used as a factor at all in assessing the fidelity of climate models (Hausfather, 2020). Effective climate sensitivity has known limitations that it describes effective feedbacks at a certain representative timescale following a change in forcing (Rugenstein 2021), but our results here highlight another issue that EffCS can only be used if we can be confident in the asymptotic energetic balance of the model. Such confidence can arise either from a ground-up demonstration of structural energy conservation in the model (Hobbs 2016), or by running sufficiently long simulations to be empirically confident both in the pre-industrial energetic balance and in the asymptotic multi-millennial tendencies of the model following a change in climate forcing. However, such experiments are currently difficult to achieve for CMIP class models, the 5000 year simulations conducted in Rugenstein (2020) were significantly longer than any experiments conducted previously - and we find in the present study that they remain too short to have confidence in the asymptotic state.*

*Given this, this study has multiple recommendations. Firstly, a greater emphasis in climate model design and quality checking needs to be placed on structural closure of the energy budget in the climate system. Models which can demonstrate that energy is conserved in the model equations can allow confidence that the system as a whole will converge to a state of true radiative equilibrium following a perturbation, which would allow a robust calculation of $EffCS$. For models which cannot demonstrate this, longer simulations are required to be confident in the asymptotic state - but these simulations may be prohibitively time and resource consuming. Such limits could potentially be alleviated through the use of lower resolution configurations (Kuhlbrodt 2018, Shields 2012) (with the risk that such models will exhibit different feedbacks from their high resolution counterparts) or by considering analytical approaches to accelerate convergence of complex systems (Xia 2012) . However, in the short term a more practical approach may be to consider alternative climate metrics which do not require assumptions about the equilibrium state of the system. Transient Climate Response does not require assumptions about radiative flux, but it does not provide direct information on the warming expected under stabilising forcing. A possible alternative is A140 (the warming observed 140 years after a step quadrupling in CO2 concentrations (Sanderson 2020), which requires no assumption on equilibrated state - and is more informative on the warming expected under high mitigation scenarios than $EffCS$ itself (even if it is known without bias due to energetic leaks). In conclusion, the role of Effective Climate Sensitivity as a metric in assessing the response of the climate system should be reconsidered, both due to its lack of relevance to projected warming under mitigation scenarios (Knutti 2017, Frame 2006, Sanderson 2020) but also due to the fact that its derivation requires assumptions about the asymptotic state of the climate system which cannot be demonstrated in a number of Earth System Models.*

*6) Figures 1 and 2 are barely discussed. The authors should add more discussion of these figures to the results section, especially highlighting any important interpretations of the figures, or move these two figures to the Appendix.*

We consider these figures to be a key part of the discussion, but there were insufficient references. A number of explicit figure references have been added where these Figures are relevant.

*7) In the results section, the authors jump back and forth between discussing Figure 3 or Figure 4 (Page 6 lines 1 – 19), making it difficult to follow the points the authors are trying to make. The authors should consider editing this section by first discussing and interpreting Figure 3, then discussing and interpreting Figure 4.*

Thanks - this text has been reformatted

*8) The authors need to verify that the figure captions match the figures. Colors and types of lines described in the figure captions do not match what was plotted in the figure, making it difficult to interpret the figures (see the Technical Corrections related to each figure below).*

Figures have been extensively revised following reviews and clarified throughout.  Apologies for caption inaccuracies in the submitted version.

*9) Table A2 is an important table, displaying the difference between EffCS computed using various methods for the LongRunMIP simulations. The authors should consider moving Table A2 into the main part of the text. They can add a discussion of the table to the results section, highlighting why the estimates for $\Delta T_{best-est}$ and $\Delta T_{extrap}$ are similar for some models yet different for others.*

Table moved to main text, thanks for the suggestion.

*Technical Corrections:*

*Equation 1b: Constant is written as $R_{4x}$, but referred to as $R_{04x}$ in the text (page 3 line 1)*

Revised, thanks.

*Table 1: $R_n$ scaling factors are not listed in Table 1, but $S_n$ scaling factors are listed. Is there a reason why the $R_n$ scaling factors are omitted?*

Now included, thanks

*Table 1: $R_0$ is included in the table, but this variable does not appear in either Eq. 1a or Eq. 1b. How does this variable relate to these two equations?*

A typo using notation from a former version.  Corrected, thanks.

*Why are the lines in figures 1 and 2 labeled as $S_{LR}$, $S_{eff}$, and $S_{extrap}$. In Eq. 1a, 1b, 2, and 3, S refers to a scaling factor. Why are the authors using this variable (S) to label the different lines?*

Notation from a former version.  Corrected, thanks.

*Figure 1 Caption:*

- *Authors state solid yellow lines are linear regressions used to estimate effective climate sensitivity for the first 150 years of data. This should be the dotted yellow lines.*

Thanks, corrected.

- *Authors state solid pink lines are linear regressions used to estimate effective climate sensitivity for the last 15% of warming. This should be the dotted pink lines.*

Thanks, corrected.

- *Authors state vertical dotted pink and yellow lines show corresponding values of effective climate sensitivity. Should be vertical solid pink and yellow lines.*

Thanks, corrected.

- *Authors state solid yellow horizonal line shows the PICTRL net energy imbalance averaged over the final 100 years of the simulation. There are no solid yellow horizontal lines. There are green horizontal lines, which are not included in the caption or legend. Are the green lines supposed to be the PICTRL net energy imbalance? If not, make sure to label what the green lines are showing.*

    Legend added.
- *Solid blue line is not described in the caption*

    Added.
- *I am not sure that the dashed blue line is described correctly in the figure caption. Authors say the dashed blue line shows an exponential model fit, but the lines in all of the subplots in Figure 1 are horizontal. Is the solid blue line actually showing the exponential model fit? If so, what do the dashed blue lines represent?*

    Sentence rewritten, legend entry added.
- *Green dots are not described in the caption*

    added

*Figure 1 General Comments:*

- *Green and blue dots in the legend representing PICTRL and ABRUPT4X are very faint, almost impossible to see. Make them more legible in legend.*

Removed from legend (high transparency points do not show well in legend, but are necessary to see the shape of the point distribution). Description remains in caption.

- *I cannot distinguish the difference between the blue dots representing ABRUPT4X and the light blue ellipse showing the 5-95 CI for $\Delta T_{extrap}$. It looks like only the light blue ellipse is plotted.*

In the longrunmip fits, the 5-95 CI is negligible - the ellipse is only visible on the CMIP5/CMIP6 fits in the supplementary. We've removed the legend entry to avoid confusion.

- *What does $n_{yr}$ show? I assume it is the number of years in the LongRunMIP simulation, but the authors should include a description of the parameter in the figure caption for clarity.*

Text removed

- *Make sure the lines plotted on the figure do not go through the text (i.e., CNRMCM61 panel has solid blue and dotted yellow lines going through $n_{yr} = 1850$)*

Text removed.

*Figure 2 Caption:*

- *There is a description of black points, but there are no black points in the figure or legend*

Corrected

- *Which dashed horizonal line illustrates $\Delta T_{extrap}$? Blue? Green?*

Corrected

- *A description of the green dashed line does not appear in the figure caption, and the green dashed line is not included in the legend.*

Corrected

- *A description of the green dots does not appear in the figure caption*

Corrected

- *Authors state the dashed purple line is $\Delta T_{best-est}$. I do not see a purple line. There is solid pink line. Is this pink line supposed to be $\Delta T_{best-est}$?*

Corrected. Yes, pink line.

*Figure 2 General Comments:*

- *Missing "of" in the sentence: "Shaded regions and thin dotted lines show the 10th and 90th percentiles of the fitted ensemble projections"*

Added.

- *The 4xCO$_2$ and pictrl is written differently from PICTRL and ABRUPT4X in the first figure caption and the main text. These scenarios should be referred to in a consistent manner*

Corrected

- *There are no lines or symbols next to 4xCO$_2$ and pictrl in the legend*

Removed (description now in caption).

*Figure 3 General Comments:*

- *It is difficult to distinguish the blue dots and the blue shaded region, specially towards the right side of each panel. Making the shaded region a different color, or different shade of blue could help distinguish the points from the shaded region.*

  Point Color changed to grey

- *Why do some of the models have visible 10th and 90th percentiles at the beginning and ending of the blue line, but others do not? What is different in the models with very small ranges of uncertainty from those with larger ranges?*

Shorter simulations (of *only* 1000 years) are subject to larger uncertainties in the fit for timescales >1000 years. This is evident for MPIESM12 and ECHAM5MPIOM. This feature is now noted in the first paragraph of the results.

- *4xCO$_2$ in the legend does not match ABRUPT4X labeling in figure caption and the main text*

corrected.

- *Missing "of" in the sentence: "Shaded regions and thin lines show the 10th and 90th percentiles of the fitted ensemble projections"*

corrected.

*Figure 4 Caption:*

- *Left hand column:*
  - *Caption says there are whiskers in the left-hand column on the light blue diamond symbols. There are no whiskers plotted showing the 10th & 90th percentiles of $\Delta T_{extrap}$*

corrected

- *Central Column:*
  - *Caption says there are cyan error bars plotted, but they are not on the figure. Only show blue diamonds*

corrected

  - *Solid and dashed yellow lines are not described in the figure caption*

Subfigure deleted

*Figure 4 General Comments:*

- *Is there any range of uncertainty for the values of $\Delta T_{best-est}$ shown by the red diamonds? If so, then this uncertainty should be indicated on the figure*

No, not as described in Rugenstein 2020.

- *There is no legend included with this figure, whereas the other 3 figures included legends. Consider adding a legend to this figure.*

Done

*Page 3 Line 31: What does "this estimate" refer to? $\Delta T_{extrap}$, $R_{extrap4x}$, or both?*

Both - now clarified in text

*Page 3 Line 35: Some other models should be included as described as behaving as expected. GISSE2R and GFDLESM2M show near zero equilibrium TOA balance in both PICTRL and ABRUPT4X simulation in Figure 3. Why were these models excluded from this sentence?*

Added, thanks.

*Why are the values in the brackets for $\Delta T_{extrap}$ and $\zeta_{extratp}$ the same in Tables A2, A3, and A4? The table caption explains that the numbers in the brackets represent the 5th and 95th percentiles. I find it highly unlikely that the 5th and 95th percentiles are the same, especially since the median value is larger than the values in the brackets.*

There was an typo in the code creating this table - 95th percentile is now correctly printed

*Page 9 Line 21: Missing closing parentheses after Table A1*

Fixed, thanks.

**References**

Hausfather, Zeke, Henri F. Drake, Tristan Abbott, and Gavin A. Schmidt. "Evaluating the performance of past climate model projections." Geophysical Research Letters 47, no. 1 (2020): e2019GL085378.

Proistosescu, C., & Huybers, P. J. (2017). Slow climate mode reconciles historical and model-based estimates of climate sensitivity. Science advances, 3(7), e1602821.

Rugenstein, Maria AA, and Kyle C. Armour. "Three flavors of radiative feedbacks and their implications for estimating equilibrium climate sensitivity." Geophysical Research Letters 48, no. 15 (2021): e2021GL092983.

Caldeira, K., and N. P. Myhrvold. "Projections of the pace of warming following an abrupt increase in atmospheric carbon dioxide concentration." Environmental Research Letters 8, no. 3 (2013): 034039.

Smith, C. J., Forster, P. M., Allen, M., Leach, N., Millar, R. J., Passerello, G. A., & Regayre, L. A. (2018). FAIR v1. 3: a simple emissions-based impulse response and carbon cycle model. Geoscientific Model Development, 11(6), 2273-2297.

Meehl, Gerald A., Catherine A. Senior, Veronika Eyring, Gregory Flato, Jean-Francois Lamarque, Ronald J. Stouffer, Karl E. Taylor, and Manuel Schlund. "Context for interpreting equilibrium climate sensitivity and transient climate response from the CMIP6 Earth system models." Science Advances 6, no. 26 (2020): eaba1981.

Jonko, Alexandra K., Karen M. Shell, Benjamin M. Sanderson, and Gokhan Danabasoglu. "Climate feedbacks in CCSM3 under changing $CO_2$ forcing. Part II: variation of climate feedbacks and sensitivity with forcing." Journal of Climate 26, no. 9 (2013): 2784-2795.

Rugenstein, Maria, Jonah Bloch-Johnson, Ayako Abe-Ouchi, Timothy Andrews, Urs Beyerle, Long Cao, Tarun Chadha et al. "LongRunMIP: motivation and design for a large collection of millennial-length AOGCM simulations." Bulletin of the American Meteorological Society 100, no. 12 (2019): 2551-2570.

Hobbs, Will, Matthew D. Palmer, and Didier Monselesan. "An energy conservation analysis of ocean drift in the CMIP5 global coupled models." Journal of Climate 29, no. 5 (2016): 1639-1653.

Kuhlbrodt, Till, Colin G. Jones, Alistair Sellar, Dave Storkey, Ed Blockley, Marc Stringer, Richard Hill et al. "The low-resolution version of HadGEM3 GC3. 1: Development and evaluation for global climate." Journal of Advances in Modeling Earth Systems 10, no. 11 (2018): 2865-2888.

Shields, C.A., Bailey, D.A., Danabasoglu, G., Jochum, M., Kiehl, J.T., Levis, S. and Park, S., 2012. The low-resolution CCSM4. Journal of Climate, 25(12), pp.3993-4014.

Sanderson, B., 2020. Relating climate sensitivity indices to projection uncertainty. Earth System Dynamics, 11(3), pp.721-735.

---

## Author Comment (AC2)

Response to Reviewer 2:

*General Comments:*

*In the manuscript under discussion, the assumption that global climate models evolve to some top-of-atmosphere radiative balance is put to the test. For this, millenia-long runs from LongrunMIP are used, alongside a linear response model with responses on three time scales. Based on the found energetic imbalances in some models and equilibrium temperature estimates, the biases in the latter are related to the former, concluding that energy leaks might influence common equilibrium climate sensitivity estimates much.*

*I find this an interesting and important exercise, with conclusions that could have big consequences for long-term projections with some global climate models. However, I am not fully convinced by the used methodology. Further, I think the text could be clearer at certain points. Finally, the presentation of the figures feels a bit sloppy with especially colors and line styles not matching with the captions. These issues should be resolved before I would recommend publication.*

Many thanks for the review. Issues with the figures and captions have been revised following comments from the reviewers, and we have taken efforts to clarify both the text and equations.

*Specific Comments:*

*(1) Central in the manuscript is the linear response type model in equations (1a)-(1b). I do not think that these equations are explained well enough nor that made choices are acknowledged and defended well enough. I also have some problems with their use for non-constant forcings.*

Thanks for this point. We have made efforts in revision to make the structural and parametric choices in this model more clear. We have also reformulated the presentation of 1b to be more intuitive to the reader. We have also revised the treatment of non-constant forcings following the reviewer's suggestion, and assessed the sensitivity of results to subjective timescale prior assumptions.

*(1a) First of all, the form of (1a)-(1b) is now defended as consistent with some simple (linear) climate models. However, it also fits with linear response theory as the response of a non-linear model "in the linear response regime". In [Proistosescu and Huybers (2017)], they already frame it in this way, and e.g. in my recent paper [Bastiaansen et al (2021)] this link is made even more explicitly. I think it would be good to clarify these things, which also would further communicate the validity of (1a)-(1b). Further, nowhere is it mentioned that equations (1a)-(1b) only hold for constant forcings, and that the parameters would be different for other forcing levels. These important 'terms and conditions' for the use of (1a)-(1b) should be added.*

These are very good points - thank you.  We make them clear in revision.

*(1b) It is now assumed that all climate models have a response on three distinct time scales. This choice for the number of time scales should be stated explicitly and a better justification needs to be given. Why should all models have the same number of response time scales? Why should there be precisely three time scales? For me, this now seemingly arbitrarily made choice is one of the weakest points of the paper and could render all your conclusions moot: what if a system actually has more than three time scales and all the remaining observed radiative imbalance disappears if you were to take all these time scales into account? So, did you check if results remain similar when a different number of time scales are used?*

Thanks for this point.  In revision, we have tested the model for a range of allowable timescales from 2 to 5 modes, assessing the residual fits as a function of timescales.

In the new scheme, we allow for timescales corresponding to subdecadal (1-10 years), decadal (10-100 years), centennial (100-1000 years), (1000-5000 years) and >5000 years. Incremental solutions allow increasingly long timescales to be considered in the fit.

 This extension has added some nuances to the conclusions of our original submission. The Figure shown here illustrates in red the number of timescales which results in the lowest overall error in fit of global mean NET TOA radiation. It is evident that some models have improved fits when the longer timescales are allowed (eg GFDLCM3).  Models describable with 3 modes (such as CCSM3) show near identical results with 3-5 timescales. Given this, we allow 5 timescales for all models for the rest of the study (which is more time consuming for the MCMC fit).

[Figure]

 However, the longrunmip simulations are insufficiently long (~5000 years) to constrain the exponential decay with those models which exhibit decays on a timescale of >5000 years. Thus, for these models, we now have larger uncertainty bars in our estimates of equilibrium response for some models (e.g. echam4mpiom).

After careful assessment, we think that this uncertainty is a real reflection of the family of plausible evolution on a multi-millennial timescale which is unconstrained by the Longrunmip simulations as they stand.  Notably - in the plot below, those models where the >5000yr timescale is well constrained (e.g. GISSE2R), there is a visible tendency towards a stable TOA balance in the LongrunMIP simulation which helps to rule out responses on this

longer timescale. For a model where the long timescale is not well constrained (e.g. ECHAM5MPIOM), there is a residual near-linear trend in TOA fluxes over the 5000 year simulation which would be consistent with a wide range of possible exponential decay fits.

We must also consider the possibility for these models that there is no stable state. If energy leaks are a function of the climate state, and the system is not tending towards a state of radiative equilibrium, our evidence that models are converging to a stable temperature is empirical - and longer simulations will be required to investigate these multi-millennial dynamics. Unfortunately, for these models - it means we can say less about their long term trajectory and equilibrium state following the 4xCO2 perturbation, which will now be one of the conclusions of our study.

[Figure]

*(1c) For a few models in LongrunMIP, the abrupt4xCO2 experiment was not long enough, and the results for a different forcing scenario were added at the end of the abrupt4xCO2 simulation in an attempt to construct a long enough simulation. However, the used linear response model in equations (1a)-(1b) is only valid for constant forcings, but the used runs have non-constant forcings (1pct2x, 1pct4x and RCP8.5). To me, that means the equations simply cannot be used. In particular, the timing of forcing in these experiments is of uttermost importance to properly assess the response over time, and splicing runs together like this therefore makes no sense to me. An alternative would be to derive a linear response model for the used forcing scenarios, and use that to fit the parameters from which the abrupt response could be inferred (taking some liberty with the 'ensemble-average' assumptions underlying linear response theory).*

Thanks for this.  We agree with the reviewer and have now re-done the analysis as suggested.  1pct2x and 1pct4x are fitted using standard impulse response model assumptions with a linearly increasing forcing.  For RCP8.5, we use the published forcing time series from Meinshausen 2011 (noting that this is an approximation, but the trade-offs between different forcer uncertainties are not the focus here).  Performance in producing the transient evolution of the LRMIP simulations is generally good (see attached plot).  The fitted impulse response model distributions are then used to simulate ABRUPT4X responses for the Gregory plots.

[Figure]

*(2) In (1a) and (1b) the parameters T0 and R4x are playing similar roles. However, they are not determined in the same way, as T0 is derived from the control experiment instead of fitted with the abrupt4xCO2 experiment. The reason for doing this should be explained.*

We have improved the explanation of this derivation, and simplified the equations for the paper . T0 is simply the best estimate of the pre-industrial temperature.  We now have R0 as the pre-industrial NET TOA balance, and R4x$_{extrap}$ as the equilibrium imbalance in the 4xCO2 experiment (following the reviewer's suggestion in point 4).

*(3) To obtain the model parameters from the data, one way or another a nonlinear fitting procedure needs to be used. Those can be sensitive to the choices for metaparameters -- in*

*this case, the choices for the priors (i.e. the mentioned distributions in Table 1). Did the authors check to make sure the presented results do not depend too much on these priors? Additionally, the choices for the prios should also be explained better; now, it just seems to be some made up numbers, but there certainly is some sort of rationale behind them?*

Some level of subjectivity is unavoidable in MCMC fitting, but our choices are informed by similar studies  (e.g.  Proistosescu, C., & Huybers, P. J. (2017), Rugenstein and Armour 2021, Caldeira and Myhrvold (2013),Smith 2018)  - with a preference towards broad priors.  We bin the allowable response timescales into regimes (subdecadal, decadal etc.) to allow for characterization of the response timescales of the different models.  We've now explained this in the text

*(4) Part of the goal of the paper seems to be to estimate the 'equilibrium' imbalance for abrupt4xCO2 experiments. Why would we want to use equations (1a) and (1b) for that? If one is only interested in that long-term imbalance, why would you not fit a decaying exponential to the last part of the transient of the imbalance instead? In any way, such kinds of choices should be addressed more explicitly in the text, including the rationale of making these choices.*

We have now formulated $R4x_{extrap}$ (the equilibrium TOA balance state following the abrupt4x perturbation) to be a parameter in the fitted equation as suggested.  We agree that this is a more accurate way of presenting our approach (it makes no difference to the actual derived fits, which were effectively always derived using  $R4x_{extrap}$).

$$\quad T(t) = \sum_{n=1}^{3} S_n(1 - e^{-(t/\tau_n)}) + T_0$$

$$R(t) = \sum_{n=1}^{3} R_n(-e^{-(t/\tau_n)}) + R_{extrap}^{4x}$$

*(5) Figures and captions are not in line with each other. For instance, in Figure 1, the caption talks about a yellow horizontal line but in the figures it seems to be a green horizontal line, regression lines are said to be solid lines but they appear to be dotted lines and vertical lines are said to be dotted but they appear to be solid. There are also blue lines, not all of which seem to be explained in the caption. The authors should verify that the captions match with the figures and explain all lines.*

Thanks, we have extensively revised the figures in preparing this response. We will endeavor to avoid labeling errors in the revision.

*(6) For me, one of the questions remaining after having read the text is what we should consider an equilibrium of the climate system. Would that just be the long-term response of the system, or do we actually want the system to have achieved radiative balance in some way? Most of the equilibrium climate sensitivity methods, including EffCS in the text, are basing their estimation technique on the requirement that there is radiative balance in equilibrium. However, the equations (1a)-(1b) explicitly do not require this. So for instance, the text on page 6, lines 43-44 stating that "if we do not know what the radiative imbalance will be when temperatures stabilise in an ABRUPT4X simulation, we in turn cannot predict the climate sensitivity with precision", hinges on what we interpret as equilibrium; in fact, you could argue that the method used in this paper is an example of a climate sensitivity prediction that does not require prior knowledge on the radiative imbalance in equilibrium, making this statement in the discussion incorrect with regard to the rest of the text. But above all, I think all these points relate to what we define as equilibrium: Originally we would say that it refers to a state in which there is radiative balance. Then when we found consistent imbalance even in the control simulation, we redefined equilibrium to mean having an imbalance similar to the control simulation. And now this paper seems to shake up even that definition in some models. In short, I think the paper would benefit from a discussion of the definition of an equilibrium climate state, relating it to the radiative imbalance and incorporating the implications of the current paper.*

This is a good point. Nomenclature has been an issue since the outset of this study. It is an unavoidable conclusion of the study that some models do not conserve energy, and that those energy leaks have the potential to be functions of climate forcing. As such, models with such leaks will not reach true energetic equilibrium as defined by dN/dt=0. This still allows for the model to reach an asymptotic stable state, which includes the energy leak - but it does not allow for the derivation of effective climate sensitivity which requires prior knowledge of the asymptotic equilibrium TOA balance. The method suggested here presents an alternative approach for deriving climate sensitivity, but it is clearly less than ideal - requiring simulations of 5000 years of simulation (or greater, as we learn in revision) to produce a robust estimate of the equilibrium state.

*(7) For me, the discussion is missing some sort of recommendation or directive to follow-up on the found energy balance issue. I know that on page 6, lines 45-51 there is some text on this, but I feel like it could be a bit more concrete. Should we conclude from this paper that estimates based on inferred equilibrium radiative (im)balance are inapt for some (or all) global climate models? And should we therefore not use such estimation methods anymore? Should we conclude from the paper that global climate models have energy leaks? And should we therefore not trust these on long time scales? Should we make sure that our global climate models have no energy leaks, and e.g. move towards climate models that are discretized in a way that prevents energy leaks or prevent energy leaks by going to*

*finer resolutions? I don't expect the authors to answer all these questions, but at least posing some of these might give the paper some more direction and might make the implications of their findings more clear.*

Thanks for this point. We copy below our expanded discussion on future recommendations for the community following this study:

*This directly impacts our ability to accurately measure $EffCS$ from short simulations, and draws into question whether $EffCS$ should be used as a factor at all in assessing the fidelity of climate models (Hausfather, 2020). Effective climate sensitivity has known limitations that it describes effective feedbacks at a certain representative timescale following a change in forcing (Rugenstein 2021), but our results here highlight another issue that EffCS can only be used if we can be confident in the asymptotic energetic balance of the model. Such confidence can arise either from a ground-up demonstration of structural energy conservation in the model (Hobbs 2016), or by running sufficiently long simulations to be empirically confident both in the pre-industrial energetic balance and in the asymptotic multi-millennial tendencies of the model following a change in climate forcing. However, such experiments are currently difficult to achieve for CMIP class models, the 5000 year simulations conducted in Rugenstein (2020) were significantly longer than any experiments conducted previously - and we find in the present study that they remain too short to have confidence in the asymptotic state.*

*Given this, this study has multiple recommendations. Firstly, a greater emphasis in climate model design and quality checking needs to be placed on structural closure of the energy budget in the climate system. Models which can demonstrate that energy is conserved in the model equations can allow confidence that the system as a whole will converge to a state of true radiative equilibrium following a perturbation, which would allow a robust calculation of $EffCS$. For models which cannot demonstrate this, longer simulations are required to be confident in the asymptotic state - but these simulations may be prohibitively time and resource consuming. Such limits could potentially be alleviated through the use of lower resolution configurations (Kuhlbrodt 2018, Shields 2012) (with the risk that such models will exhibit different feedbacks from their high resolution counterparts) or by considering analytical approaches to accelerate convergence of complex systems (Xia 2012) . However, in the short term a more practical approach may be to consider alternative climate metrics which do not require assumptions about the equilibrium state of the system. Transient Climate Response does not require assumptions about radiative flux, but it does not provide direct information on the warming expected under stabilising forcing. A possible alternative is A140 (the warming observed 140 years after a step quadrupling in CO2 concentrations (Sanderson 2020), which requires no assumption on equilibrated state - and is more informative on the warming expected under high mitigation scenarios than $EffCS$ itself (even if it is known without bias due to energetic leaks). In conclusion, the role of Effective Climate Sensitivity as a metric in assessing the response of the climate system should be reconsidered, both due to its lack of relevance to projected warming under mitigation scenarios (Knutti 2017, Frame 2006, Sanderson 2020) but*

*also due to the fact that its derivation requires assumptions about the asymptotic state of the climate system which cannot be demonstrated in a number of Earth System Models.*

*Technical Corrections:*

*(1) The top-of-atmosphere radiative balance in Figure 1 and Figure 3 have different signs. That should be made consistent.*

Thanks, fixed following the standard sign convention used in Gregory plots.

*(2) In section 2.1, it is said that Table 1 contains parameter ranges and constraints. The caption does, however, say these are the prior ranges. Could the authors confirm that the values in Table 1 indeed correspond to the prior values and that the parameter ranges are not constrained in the fitting procedure? If so, also the text in section 2.1 should be changed to reflect that.*

Thanks, text updated to confirm that these are the prior ranges.

*(3) Table 1 misses the parameters R_n.*

Not used now (see response to main point 2)

*(4) Figure 4: the whiskers in the sensitivity and zeta plots are missing.*

This was a plot resolution issue, fixed now.

*(5) In the tables at the end of the paper, I was not able to find the fitted values for Sn, Rn and especially tau_n.*

Thanks - this was not in a table.  We include a new table with mean MCMC parameters in revision:

| Model | $S_1$ | $S_2$ | $S_3$ | $S_4$ | $S_5$ | $R_1$ | $R_2$ | $R_3$ | $R_4$ | $R_5$ | $R\tau_1$ | $\tau_2$ | $\tau_3$ | $\tau_4$ | $\tau_5$ | $R_{extrap}^{4x}$ |
|---|---|---|---|---|---|---|---|---|---|---|---|---|---|---|---|---|
| CCSM3 | 2.48 | 1.11 | 1.49 | 0.03 | 0.04 | 8.49 | 1.50 | 1.32 | 0.16 | 0.26 | 0.96 | 22.99 | 514.18 | 2980.28 | 7585.31 | -0.10 |
| CESM104 | 3.18 | 1.60 | 1.16 | 0.35 | 0.88 | 4.16 | 1.23 | 0.93 | 0.29 | 0.45 | 4.76 | 80.11 | 350.89 | 3322.39 | 7609.84 | -0.22 |
| CNRMCM61 | 2.65 | 2.74 | 3.36 | 0.26 | 0.41 | 3.43 | 1.38 | 1.49 | 0.30 | 0.50 | 3.23 | 36.04 | 318.87 | 3118.36 | 7608.10 | 1.96 |
| ECEARTH | 0.85 | 4.04 | 2.35 | 0.36 | 0.70 | 3.22 | 1.46 | 2.32 | 0.01 | 0.01 | 7.58 | 43.09 | 269.63 | 2612.42 | 7503.20 | -0.63 |
| ECHAM5MPIOM | 1.66 | 4.06 | 4.48 | 0.94 | 1.63 | 3.41 | 1.92 | 1.18 | 1.82 | 3.02 | 3.01 | 38.34 | 430.45 | 3180.65 | 7611.00 | -1.54 |
| FAMOUS | 3.02 | 5.58 | 3.52 | 1.43 | 0.93 | 3.91 | 1.96 | 0.38 | 0.05 | 0.08 | 5.18 | 35.91 | 323.58 | 3121.48 | 7732.14 | 0.18 |
| GFDLCM3 | 0.57 | 2.33 | 5.68 | 1.31 | 0.69 | 5.05 | 1.60 | 1.71 | 0.90 | 0.57 | 5.32 | 69.16 | 721.86 | 2498.57 | 7620.66 | -0.32 |
| GFDLESM2M | 2.12 | 0.05 | 3.95 | 0.62 | 0.11 | 5.07 | 0.77 | 2.24 | 0.19 | 0.12 | 4.28 | 42.25 | 406.82 | 2601.13 | 7417.61 | -0.24 |
| GISSE2R | 2.44 | 0.94 | 1.40 | 0.02 | 0.01 | 5.37 | 1.36 | 1.44 | 0.07 | 0.05 | 2.12 | 88.29 | 680.85 | 2704.30 | 7642.74 | 0.07 |
| HadCM3L | 2.58 | 1.59 | 1.60 | 0.45 | 0.91 | 3.79 | 0.55 | 1.08 | 0.63 | 1.37 | 4.91 | 45.96 | 245.51 | 3201.77 | 7731.69 | -0.85 |
| HadGEM2 | 4.11 | 1.29 | 1.83 | 2.00 | 3.86 | 4.54 | 2.28 | 1.28 | 0.79 | 1.76 | 1.53 | 14.19 | 247.97 | 3372.98 | 7536.63 | -1.01 |
| IPSLCM5A | 2.37 | 2.59 | 2.22 | 0.37 | 0.75 | 2.19 | 1.51 | 1.10 | 0.62 | 1.11 | 5.39 | 33.86 | 238.98 | 3230.48 | 7566.52 | -0.25 |
| MIROC32 | 4.15 | 1.32 | 3.03 | 0.07 | 0.10 | 4.28 | 1.74 | 2.01 | 0.22 | 0.28 | 4.46 | 17.80 | 585.56 | 3117.25 | 7697.05 | 1.02 |
| MPIESM11 | 2.43 | 1.51 | 1.68 | 1.30 | 0.06 | 4.14 | 2.05 | 1.38 | 0.95 | 0.20 | 3.42 | 28.18 | 254.16 | 1490.62 | 7421.48 | 0.31 |
| MPIESM12 | 2.56 | 1.56 | 1.86 | 0.71 | 1.52 | 4.42 | 1.96 | 1.50 | 0.84 | 1.68 | 2.85 | 20.90 | 272.32 | 3211.09 | 7750.07 | -1.21 |

*(6) The tables at the end of the paper, the 95% values seem to just state the 5% values again for some of the parameters.*

Thanks, fixed.

**References**

Proistosescu, C., & Huybers, P. J. (2017). Slow climate mode reconciles historical and model-based estimates of climate sensitivity. Science advances, 3(7), e1602821.

Rugenstein, Maria AA, and Kyle C. Armour. "Three flavors of radiative feedbacks and their implications for estimating equilibrium climate sensitivity." Geophysical Research Letters 48, no. 15 (2021): e2021GL092983.

Caldeira, K., and N. P. Myhrvold. "Projections of the pace of warming following an abrupt increase in atmospheric carbon dioxide concentration." Environmental Research Letters 8, no. 3 (2013): 034039.

Smith, C. J., Forster, P. M., Allen, M., Leach, N., Millar, R. J., Passerello, G. A., & Regayre, L. A. (2018). FAIR v1. 3: a simple emissions-based impulse response and carbon cycle model. Geoscientific Model Development, 11(6), 2273-2297.

Meehl, Gerald A., Catherine A. Senior, Veronika Eyring, Gregory Flato, Jean-Francois Lamarque, Ronald J. Stouffer, Karl E. Taylor, and Manuel Schlund. "Context for interpreting equilibrium climate sensitivity and transient climate response from the CMIP6 Earth system models." Science Advances 6, no. 26 (2020): eaba1981.

Jonko, Alexandra K., Karen M. Shell, Benjamin M. Sanderson, and Gokhan Danabasoglu. "Climate feedbacks in CCSM3 under changing CO 2 forcing. Part II: variation of climate feedbacks and sensitivity with forcing." Journal of Climate 26, no. 9 (2013): 2784-2795.

Rugenstein, Maria, Jonah Bloch-Johnson, Ayako Abe-Ouchi, Timothy Andrews, Urs Beyerle, Long Cao, Tarun Chadha et al. "LongRunMIP: motivation and design for a large collection of millennial-length AOGCM simulations." Bulletin of the American Meteorological Society 100, no. 12 (2019): 2551-2570.

Hobbs, Will, Matthew D. Palmer, and Didier Monselesan. "An energy conservation analysis of ocean drift in the CMIP5 global coupled models." Journal of Climate 29, no. 5 (2016): 1639-1653.

Kuhlbrodt, Till, Colin G. Jones, Alistair Sellar, Dave Storkey, Ed Blockley, Marc Stringer, Richard Hill et al. "The low‑resolution version of HadGEM3 GC3. 1: Development and evaluation for global climate." Journal of Advances in Modeling Earth Systems 10, no. 11 (2018): 2865-2888.

Shields, C.A., Bailey, D.A., Danabasoglu, G., Jochum, M., Kiehl, J.T., Levis, S. and Park, S., 2012. The low-resolution CCSM4. Journal of Climate, 25(12), pp.3993-4014.

Sanderson, B., 2020. Relating climate sensitivity indices to projection uncertainty. Earth System Dynamics, 11(3), pp.721-735.

Meinshausen, M., S. J. Smith, K. V. Calvin, J. S. Daniel, M. L. T. Kainuma, J.-F. Lamarque, K. Matsumoto, S. A. Montzka, S. C. B. Raper, K. Riahi, A. M. Thomson, G. J. M. Velders and D. van Vuuren (2011). "The RCP Greenhouse Gas Concentrations and their Extension from 1765 to 2300." Climatic Change (Special Issue), DOI: 10.1007/s10584-011-0156-z, freely available online

---

## Author Response (AR2)

**Minor revisions to EGUSPHERE-2022-167**

Many thanks to the reviewers for their input during the publication of this paper.  We are pleased to address the remaining concerns in this revision.

**Reviewer 1**

1) Referring to Table 3 before Table 2 has been discussed or shown comes across weirdly when read (Page 3 line 26). Can the authors reorder the tables, so they are referred to in chronological order?

*Done - order reversed*

2) Figure 1 Caption: "The shortest timescale model with errors within a .5 percent" should be "0.5 percent".
*Done*

3) Figure 2 Caption: The blue line does not appear on every panel. The authors should consider making a note in the caption that the blue line does not appear for every model, and why.
*Done*

4) Figure 2 legend: The light green dots are not included in the legend
*Now included*

5) Figure 3 legend: Grey and green dots are not included in the legend
*Now included*

6) Figure 4 legend: Green dots are not included in the legend
*Now included*

7) figure 4 caption: The authors should provide a disclaimer in the caption that the y-axis for each panel is not the same. For example, CESM104 goes to about 7.5 K, but CNRMCM61 right next to it exceeds 10 K. Figures 1, 2, and 3 have the same y-axis across all of the panels, so the reader might not notice they change in Figure 4.
*Done*

**Reviewer 2**

In the text following equations (1), it is said that these apply to a unit step change in forcing, but later it is said the ABRUPT4x simulations are directly fitted to these equations. Judging by the parameter labels in equations (1), I suspect the "unit step size" is supposed to be the abrupt 4xCO2 forcing, but this is not clarified in the text. Further, if that is the case, equations (2) are

missing a scaling factor (by the forcing in abrupt4xCO2). This should be clarified and harmonized in the text.

*This interpretation is correct.  Text and equations have been adjusted to reflect this.*